# Dual receptor-sites reveal the structural basis for hyperactivation of sodium channels by poison-dart toxin batrachotoxin

Lige Tonggu [1,7], Goragot Wisedchaisri [1,7], Tamer M. Gamal El-Din [1,7], Michael J. Lenaeus[2], Matthew M. Logan[3,5], Tatsuya Toma[3,6], Justin Du Bois [3], Ning Zheng [1,4] ✉ & William A. Catterall [1] ✉

The poison dart toxin batrachotoxin is exceptional for its high potency and toxicity, and for its multifaceted modification of the function of voltage-gated sodium channels. By using cryogenic electron microscopy, we identify two homologous, but nonidentical receptor sites that simultaneously bind two molecules of toxin, one at the interface between Domains I and IV, and the other at the interface between Domains III and IV of the cardiac sodium channel. Together, these two bound toxin molecules stabilize α/π helical conformation in the S6 segments that gate the pore, and one of the bound BTX-B molecules interacts with the crucial Lys1421 residue that is essential for sodium conductance and selectivity via an apparent water-bridged hydrogen bond. Overall, our structure provides insight into batrachotoxin's potency, efficacy, and multifaceted functional effects on voltage-gated sodium channels via a dual receptor site mechanism.

Voltage-gated sodium channels (Na$_V$s) initiate action potentials in nerve and muscle[1,2]. Na$_V$s are composed of four pseudosymmetric, homologous domains (*D*I-*D*IV) having six transmembrane segments each (S1–S6), which surround a central transmembrane pore[3,4]. Segments S1–S4 form the voltage sensors (VS) in each domain. Segments S5 and S6, and the P loop between them, form the pore module (PM). Structural studies using X-ray crystallography revealed the overall transmembrane structure of ancestral bacterial sodium channels[5–7], and analyses by cryogenic electron microscopy (cryo-EM) have elucidated the structures of sodium channels from mammalian nerve[8,9], skeletal muscle[10], and heart[11]. In all of these structures, the central PM is composed of an extracellular vestibule, a narrow ion selectivity filter, a central cavity, and an activation gate formed by the intracellular ends of the pore-lining S6 segments. The four VS surround the central PM in a square array and modulate pore opening through the connecting S4-S5 linkers. Depolarized membrane potentials activate the VS from *D*I to *D*III, which leads to pore opening and sodium influx. Within milliseconds, the VS in *D*IV triggers fast inactivation to close the pore using the fast inactivation gate located in the intracellular *D*III-*D*IV linker.

Many highly toxic small molecules and peptides produced by a wide range of plant and animal species bind to five or more specific receptor sites on sodium channels and paralyze predators and prey[4,12,13]. Batrachotoxin (BTX) is unique in that it is small, skin permeable, and highly potent, with a mean lethal dose in mice of 2 μg/kg[14], the most potent of all sodium channel toxins. BTX is found predominantly in poison dart frogs in Central and South America[15–18] and in toxic birds in Papua New Guinea[19]. The Dendrobatid frogs use BTX for defense against attacks by predators[17,18], which provides a graphic illlustration of its speed and potency. The poison secreted from the skin of these tiny frogs has been used for centuries by indigenous tribes for hunting[16–18]. A single *Phyllobates terribilis* frog can poison many darts, which are each sufficient to immobilize or kill prey.

[1]Department of Pharmacology, University of Washington, Seattle, WA 98195, USA. [2]Division of General Internal Medicine, Department of Medicine, University of Washington, Seattle, WA 98195, USA. [3]Department of Chemistry, Stanford University, Stanford, CA 94305, USA. [4]Howard Hughes Medical Institute, University of Washington, Seattle, WA 98195, USA. [5]Present address: Vividion Therapeutics, Inc., 5820 Nancy Ridge Dr., San Diego, CA 92121, USA. [6]Present address: PRISM BioLab Co., Ltd., 2-26-1 Muraokahigashi, Fujisawa-shi, Kanagawa 251-8555, Japan. [7]These authors contributed equally: Lige Tonggu, Goragot Wisedchaisri, Tamer M. Gamal El-Din. ✉e-mail: nzheng@uw.edu; wcatt@uw.edu

Structural studies show that the Neurotoxin Receptor Site I, which binds the pore-blocking toxins tetrodotoxin, saxitoxin and μ-conotoxin, is located in the outer PM[9,10], and the receptor sites for Site III and Site IV polypeptide toxins from spiders and scorpions are located on the extracellular surfaces of the VS in Domains IV and II, respectively[9,20–22]. In contrast, mutagenesis and structure/function studies indicate that Neurotoxin Receptor Site II containing the BTX binding site is located in the central cavity of the PM, spanning Domains I, III and IV, with contact residues from transmembrane segments $D$I-S6, $D$III-S6 and $D$IV-S6[23–27]. As a selective agonist, BTX causes a hyperpolarizing shift in the voltage-dependence of activation, inhibition of fast inactivation, reduced single channel conductance, and a striking ten-fold increase in calcium selectivity for Na$_V$s[28]. The negative shift in the voltage dependence of activation and the block of fast inactivation generate repetitive action potential firing and persistent depolarization, whereas the increase in calcium permeation further induces hyperactivity at synapses, hypercontraction of skeletal muscles, and life-threatening arrhythmias in the heart[29–32]. All of this storm of gain-of-function effects requires persistent activation of only a few percent of Na$_V$ channels, contributing greatly to the potency of the lethal effects of the toxin. Aconitine and veratridine from plants in the *Aconitum* and *Veratrum* genera, respectively, also act at Site II, but with lower affinity and potency than BTX[33–35]. The structural and chemical basis for the high potency of BTX, and its strong effects on voltage-dependent activation, fast inactivation, ion conductance, and calcium selectivity has remained unclear.

Here, by taking advantage of new methods for total synthesis of BTX and derivatives[36,37], we report the cryo-EM structure of the cardiac sodium channel Na$_V$1.5 in complex with the BTX derivative batrachotoxinin-A 20-α-benzoate (BTX-B), functionally equivalent to BTX[24,36–39] (Fig. 1a). Surprisingly, our results reveal dual toxin/receptor interaction sites at the Domain I/IV and Domain III/IV interfaces in the central cavity. These unexpected structural findings provide mechanistic insight into the high potency and efficacy as well as the multifaceted functional effects of BTX on Na$_V$s.

## Results

### BTX-B effects on rNa$_V$1.5c

BTX is an ester of a 20-α-pyrrole carboxylic acid with the complex steroidal alkaloid Batrachotoxinin-A (BTX-A; Fig. 1a). Batrachotoxinin-A 20-α-benzoate (BTX-B; Fig. 1b) is an ester of benzoic acid and the same complex steroidal alkaloid backbone, in which the benzoate moiety replaces the 20-α-pyrrole ester in native BTX (Fig. 1b)[38]. Importantly, BTX-B and BTX bind to the same receptor site and have equivalent potency and efficacy on nerve and muscle sodium channels[35,38]. For structural studies, we used the construct rat Na$_V$1.5c with deletions in the $D$I-$D$II and $D$II-$D$III intracellular linkers, and the C-terminus (rNa$_V$1.5c) in order to enhance expression and improve resolution in structural analysis[40]. The rNa$_V$1.5c protein was expressed in HEK293 cells as reported previously (see Methods)[40,41]. Patch-clamp electrophysiological recording showed that rNa$_V$1.5c is sensitive to BTX-B (Fig. 1c). At 10 μM concentration, BTX-B prolonged the peak current by nearly completely inhibiting fast inactivation (Fig. 1c). We used this potent effect of BTX-B as a quantitative metric of toxin activity. However, in contrast to its action on native sodium channels, BTX-B negatively shifted the foot of the activation curve but did not significantly shift the voltage-dependent activation of rNa$_V$1.5c measured at the midpoint relative to the control (Fig. 1d). This is likely because the rNa$_V$1.5c channel already has a strong negative shift in the voltage dependence of activation compared to wild type (WT)[11], which might occlude the full effect of BTX-B to further negatively shift the G/V curve.

To determine the structure of rNa$_V$1.5c in complex with BTX-B, rNa$_V$1.5c was purified in the presence of 600 nM BTX-B throughout (see Methods). Purified rNa$_V$1.5c was mixed with the α-scorpion toxin

LqhIII, which prevents fast inactivation and increases the affinity for BTX[12], and the complex was further purified by size-exclusion chromatography (Supplementary Fig. 1a, b). We performed cryo-EM single particle analysis of the rNa$_V$1.5c/BTX-B sample frozen in vitreous ice from 7542 movie stacks collected on a 300-KeV Titan Krios microscope (Supplementary Figs. 1c, d and 2a, and Supplementary Table 1). A total of 86,763 particles were selected for refinement and 3D reconstruction, which yielded the final map at 3.3 Å overall resolution according to the Gold Standard Fourier Shell Correlation (FSC) criterion of 0.143 (Supplementary Fig. 2b). In the resulting structure, the transmembrane PM has a local resolution of ~3.0 Å, while the resolution of the intracellular N-terminal domain (NTD), VS, and the extracellular loops (ECL) is between 3.0–4.0 Å (Fig. 1e). In discussing individual amino acid residues, we use their numbers in the complete rNa$_V$1.5 amino acid sequence so that the deletions of intracellular domains in the Na$_V$1.5c construct do not alter the residue numbers.

### Overall structure

The structure of rNa$_V$1.5c/BTX-B contains 1238 amino acids spanning residue 11 in the NTD to residue 1780 at the end of the $D$IV-S6 segment (Fig. 1e, f). The cryo-EM density for the NTD (residues 11–32 and 53–117) has a local resolution range of 3.5–4.0 Å (Fig. 1e) and resembles the N-terminal domain in both the predicted structure by AlphaFold[42], and the conformation revealed in two recently reported structures of human Na$_V$1.7[43–45]. The well-resolved density of the map in this region allowed unambiguous docking and incorporation of the NTD into the model of Na$_V$1.5 for the first time (Fig. 1f and Supplementary Fig. 3; Supplementary Information). The NTD appears to be stabilized by the intracellular helix of the $D$I-S6 segment that extends far beyond the membrane into the cytoplasm (Supplementary Fig. 3a, b) and by a hydrogen bond with the $D$I S4-S5 linker (Supplementary Fig. 3c). Since the NTD and the intracellular $D$I-S6 helix were not well resolved in any previously reported structures of Na$_V$1.5, we hypothesize that the binding of BTX-B may stabilize this region through allosteric interactions. As the BTX-B receptor site includes several residues in the upstream transmembrane region of the $D$I-S6 segment (see below), these interactions may help to rigidify the remaining cytoplasmic $D$I-S6 helix and in turn stabilize the conformation of the NTD.

Annular lipids/detergents are also well defined in the density map, and the final model includes putative cholesterol hemisuccinate (CHS), 1-palmitoyl-2-oleoyl-sn-glycero-3-phosphocholine (POPC), and glycodiosgenin (GDN) (Fig. 1f and Supplementary Fig. 4; Supplementary Information). CHS and GDN were included in the sample preparation, while POPC is a representative of major phospholipids co-purified from the plasma membrane. Seven N-linked glycosylation sites are located in the ECL connecting the S5 segments and P-loop segments in $D$I (Asn284, Asn319, Asn329) and $D$III (Asn1367, Asn1376, Asn1382, Asn1390) (Fig. 1f).

The overall structure of rNa$_V$1.5c/BTX-B is similar to that of apo rNa$_V$1.5c (PDB: 3UZ3) and flecainide-bound rNa$_V$1.5c (PDB: 3UZ0), with a root mean square deviation (RMSD) for Cα atoms of ~1.2 Å and ~1.0 Å, respectively, indicating essentially no major conformational changes in the backbone of rNa$_V$1.5c upon BTX-B binding. The S4 segments in all VS are translocated outward, resembling the activated VS conformation (Fig. 1f). The fast inactivation gate and its IFM motif in the $D$III-$D$IV linker are lodged in their receptor site near the junction of the $D$III and $D$IV S4-S5 linkers to keep the pore closed (Fig. 1f, orange). Despite the addition of LqhIII in our sample preparation, the toxin is not visible in the density map compared to the structure of rNa$_V$1.5c/LqhIII complex (PDB: 7K18) with the $D$IV-VS trapped in an intermediate conformation[22], possibly because the fully activated $D$IV-VS in rNa$_V$1.5c/BTX-B prevented LqhIII from binding with high affinity. Therefore, our rNa$_V$1.5/BTX-B structure is most likely captured in an inactivated state with the VS activated and the pore closed.

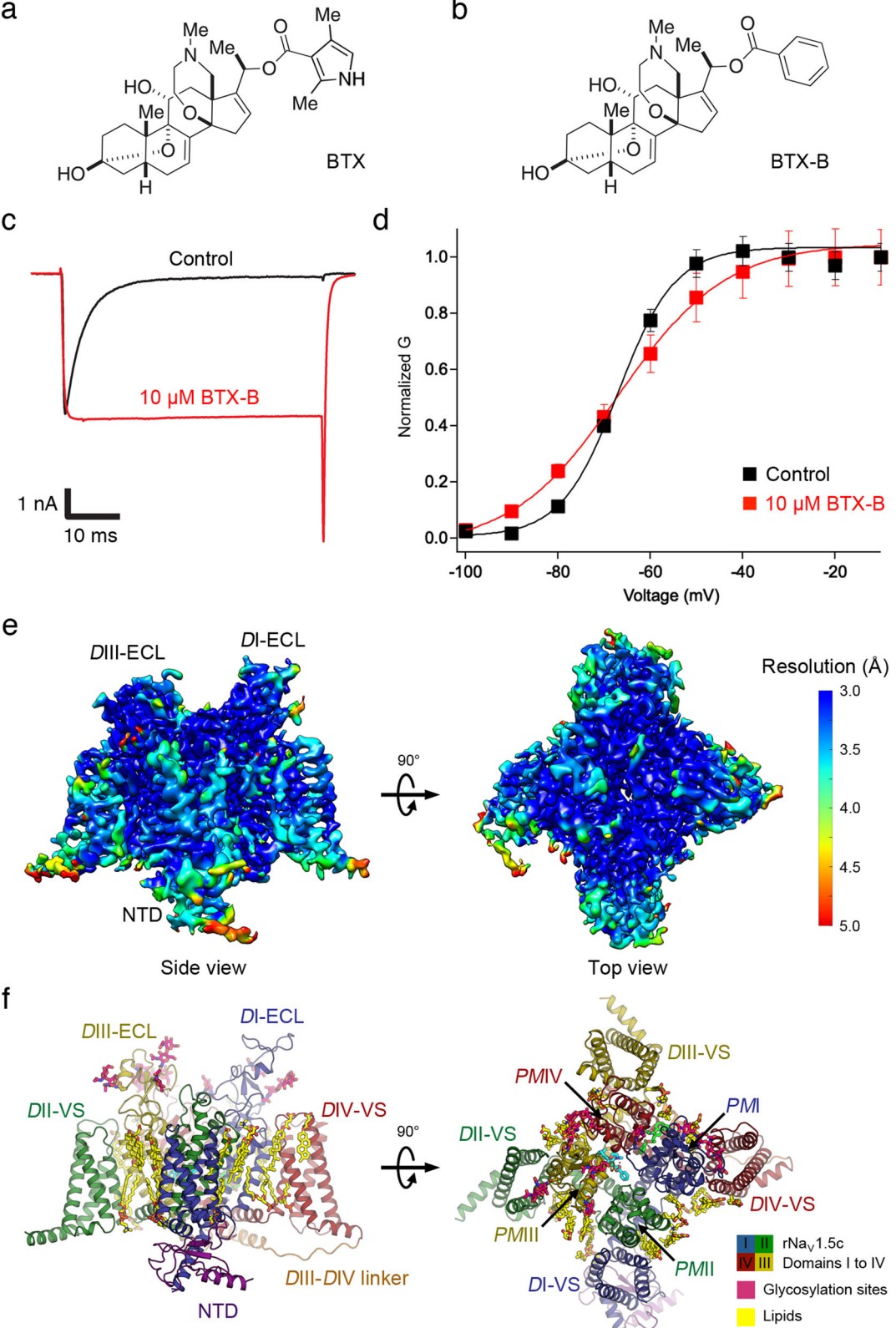

**Fig. 1 | Electrophysiological characterization and overall structure of rNa$_V$1.5c with BTX-B. a** Chemical structure of BTX. **b**. Chemical structure of BTX-B. **c** Sodium current of rNa$_V$1.5c in the absence (black) and the presence of 10 µM BTX-B (red). **d** Plots of conductance/voltage (G/V) relationship for rNa$_V$1.5c in the absence (black) and the presence of 10 µM BTX-B (red). Data are presented as mean values ± SEM with *n* = 7 cells. **e** Cryo-EM density of BTX-B bound rNa$_V$1.5c colored by local resolution (side bar). **f** Structure of BTX-B bound rNa$_V$1.5c. Each domain is represented in different colors (NTD−dark purple, *D*I−dark blue, *D*II−dark green, *D*III−dark yellow, *D*IV−dark red, *D*III-*D*IV linker−orange). Glycosylation sites and lipids are shown as sticks in fandango and bright yellow, respectively.

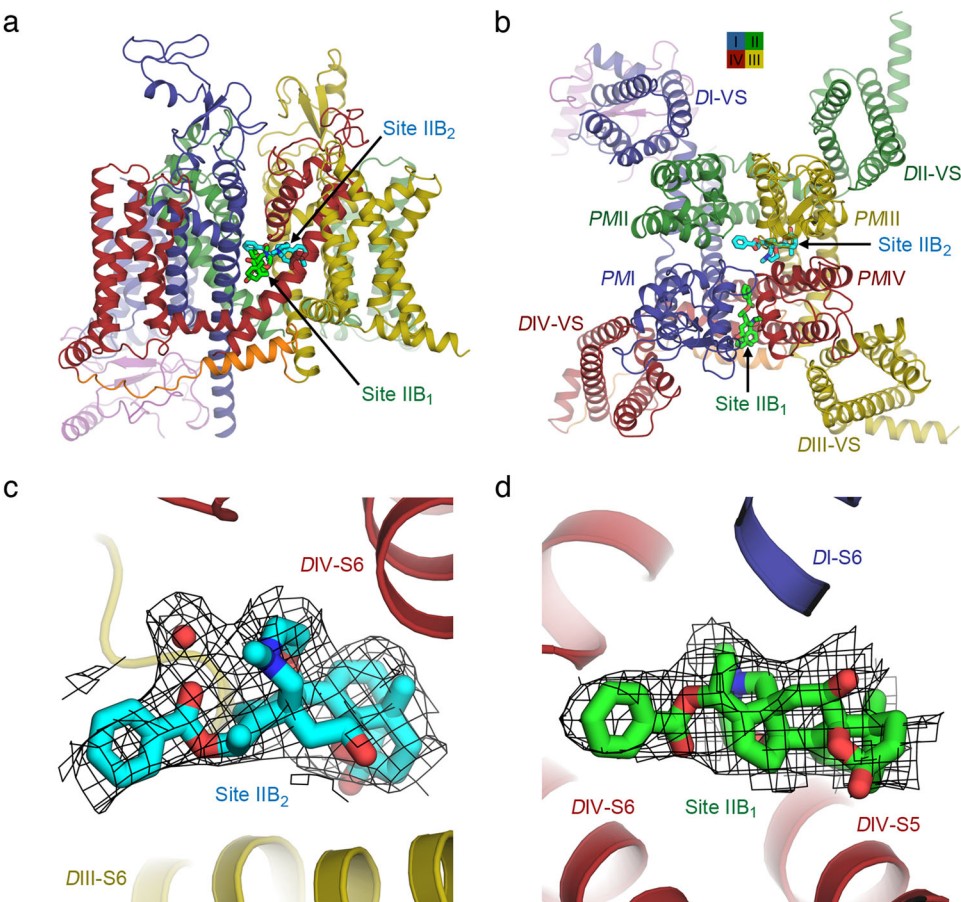

**Fig. 2 | The dual receptor sites for BTX-B in rNa_V1.5c. a** Side view of the structure of rNa_V1.5c with BTX-B Site IIB$_2$ (cyan) located in the fenestration between the *D*III (dark yellow) and *D*IV (dark red) interface, and Site IIB$_2$ (bright green) in the fenestration between the *D*I (dark blue) and *D*IV (dark red) interface. **b** Top view of the structure of rNa_V1.5c with BTX-B Sites IIB$_1$ and IIB$_2$. The two BTX-B molecules are ~7 Å apart. **c** Cryo-EM density map (black mesh, 3.5σ) of Site IIB$_2$ (cyan). **d** Cryo-EM density map (black mesh, 3.5σ) of Site IIB$_1$ (bright green).

## Dual BTX-B receptor sites

Because past mutagenesis and photolabeling studies have suggested that BTX binds in the PM near the S6 segments of Domains I, III, and IV[23–27], we examined our cryo-EM density map for strong density candidates corresponding to BTX-B in the areas that were supported by past functional studies. Two equally strong (~10σ peak) density components with a shape similar in size and form to BTX-B were observed simultaneously in the central cavity of the PM at the level of the fenestrations (Fig. 2a–d), where the local resolution of the map is ~3.0 Å. In order to distinguish these densities from endogenous lipids often found in the central cavity and the fenestrations, we compared our density map to cryo-EM maps of other related rNa_V1.5c structures, which employed similar methods for preparation and analysis but without BTX-B (Supplementary Fig. 5). The BTX-B densities are much bulkier and longer than the weaker, thinner, and shorter densities observed in other rNa_V1.5c structures without BTX-B, and the BTX-B densities resemble the BTX-B molecule in size and shape (Supplementary Fig. 5).

Because BTX-B is a lipid-soluble toxin, we expected strong hydrophobic interactions of the toxin with lipids, which may result in contiguous densities connecting the bound toxin with lipids. Indeed such contiguous density is observed in our density map (Supplementary Fig. 4), but the phospholipid is easily distinguished from BTX-B by size and shape. In addition, the density level and connectivity for adjoining lipids varied in different refinement cycles, but the strong, bulky densities for BTX-B were consistent throughout all of the refinements. Based on these observations, we therefore

concluded that the two cryo-EM densities observed in our map are bound BTX-B, with possibly some minor partial overlap of adjoining lipids, despite our best efforts to resolve these particles during image processing.

With the presence of BTX-B in our sample, the NTD is better resolved indicating some structural stabilization by BTX-B. Despite the use of LqhIII in a similar preparation as reported previously[22], which partially deactivated the *D*IV-VS into a down state when a complex is formed, the *D*IV-VS is fully activated in the presence of BTX-B. There are also clear local conformational changes (see below) in the channel to support the binding of BTX-B that result in no structural clash.

One density specific for BTX-B is located partially in the *D*III-*D*IV fenestration and on the intracellular side of the *D*III-*D*IV selectivity filter (Fig. 2a, b). The components of *PM*III and *PM*IV helices that bind BTX-B (cyan sticks) in this site are illustrated as dark yellow and dark red ribbons, respectively (Fig. 2b, c). The other density specific for BTX-B is located completely within the fenestration at the *D*I-*D*IV interface (Fig. 2a, b). BTX-B (bright green sticks) binds between the *PM*I and *PM*IV helices, as illustrated in dark blue and dark red ribbons, respectively, making strong interactions with the S6 segments in both domains (Fig. 2b, d). We designate these two BTX-B receptor sites as Site IIB$_1$ for the site in the *D*I/*D*IV interface and Site IIB$_2$ for the site in the *D*III/*D*IV interface, reflecting their functional roles as BTX-binding motifs in Neurotoxin Receptor Site II.

For Site IIB$_2$, BTX-B lies horizontally parallel to the membrane plane with the benzoate moiety facing the center of the pore, while the BTX-A core is inserted into the fenestration between the S6 segments

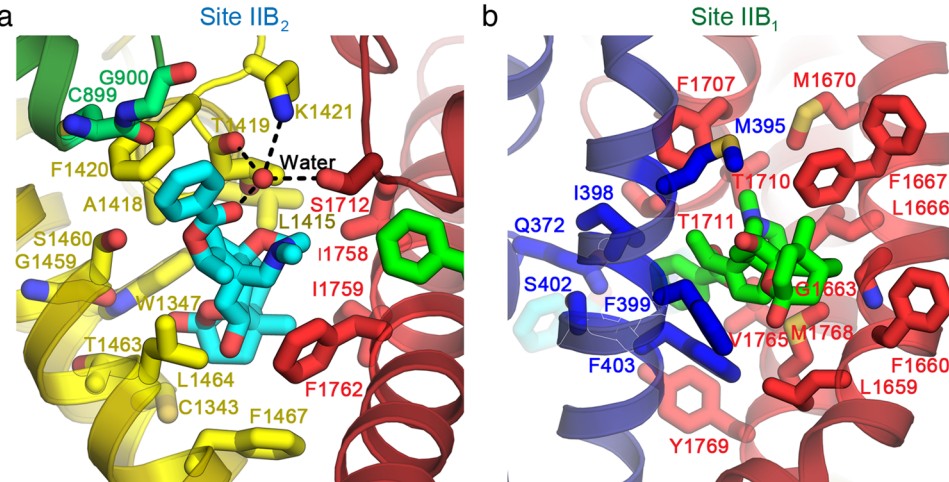

**Fig. 3 | Close up views of the dual receptor sites. a** Site IIB$_2$ (cyan) located between *D*III (dark yellow) and *D*IV (dark red) interface. Amino acid residues that form the binding site are shown as sticks in brighter colors according to their domains (yellow for *D*III and red for *D*IV). Two residues (lime green) in *D*II also contribute their main chain to the binding site. Part of BTX-B from Site IIB$_1$ (bright green) can be seen on the right. The stereo view of this receptor site with cryo-EM density map is shown in Supplementary Fig. 6a. Also see Supplementary Movie 1. **b** Close up view of the receptor site for Site IIB$_1$ (bright green) located between *D*I (dark blue) and *D*IV (dark red) interface. Amino acid residues that form the binding site are shown as sticks in brighter colors according to their domains (blue for *D*I and red for *D*IV). F399 and F403 in *D*I-S6 adopt the down rotamers in this model. In the structures of

antiarrhythmic drug flecainide- and quinidine-bound rat and human Na$_V$1.5 (PDBs: 6UZ0 and 6LQA, respectively)[11,46], F399 and F403 both adopt an "up" rotamer that would clash with BTX-B. Furthermore, the F1667 side chain from *D*IV-S5 clearly adopts two partial rotameric conformations with equal distribution, one rotamer in the "down" conformation, closing down the interaction with the BTX-A moiety of the BTX-B site IIB$_1$, whereas the other rotamer in the "up" conformation makes no interaction with the bound BTX-B (Fig. 3b, Supplementary Fig. 6b, and Supplementary Movie 2). In all of the other Na$_V$1.5 structures mentioned above, F1667 uniformly adopts an "up" rotamer that points away from the BTX-B site IIB$_1$ suggesting that BTX-B binding favors the "down" rotamer. See also Supplementary Fig. 6.

from *D*III (dark yellow) and *D*IV (dark red; Fig. 2a, b). The mesh representing cryo-EM density fits the toxin structure closely (cyan sticks, Fig. 2c). For Site IIB$_1$, BTX-B also lies horizontally parallel to the membrane plane, but completely in the fenestration between *D*I and *D*IV, with the benzoate moiety pointing toward the pore (Fig. 2a, b). The mesh representing cryo-EM density fits closely with the structure of bound BTX-B (bright green sticks, Fig. 2d). Thus, these two toxin receptor sites lie in homologous subunit interfaces in pseudosymmetric positions, poised to modulate voltage-dependent gating in Domains I, III, and IV and ion selectivity through interactions with the nearby ion selectivity filter formed by the P loops.

## Contact Amino Acid Residues in the Dual BTX-B Receptor Sites

As illustrated in Fig. 3a, the bound BTX-B at Site IIB$_2$ (cyan) nests in a cradle formed by hydrophobic residues from the *D*II P loop (green), segments *D*III-S5 and *D*III-S6 (Fig. 3a, yellow), and segment *D*IV-S6 (Fig. 3a, red; Supplementary Fig. 6a; and Supplementary Movie 1). As highlighted in yellow in Fig. 3a, the *D*III P-loop amino acid residues L1415, A1418, T1419, F1420, and the *D*III S6 residues G1459, S1460, T1463, L1464, and F1467 form the main binding site for the BTX-A core (Fig. 3a, yellow; Supplementary Fig. 6a; and Supplementary Movie 1, yellow sticks). Remarkably, K1421, which is part of the conserved DEKA locus in the *D*III P-loop that forms the selectivity filter for sodium ions, interacts with the carbonyl oxygen of the benzoate moiety of BTX-B through an unexpected water-mediated hydrogen-bonding interaction (Fig. 3a, yellow). This carbonyl oxygen in BTX-B is also present in native BTX, highlighting a critical role of this functional group in altering ion selectivity by both compounds (Fig. 1a). The main chain oxygen of T1419 from the *D*III P-loop and the side chain of S1712 from the *D*IV P-loop also coordinate this water molecule. Moreover, F1420 adjacent to K1421 forms an orthogonal (T-shape) π-π stacking with the aryl ring of the benzoate moiety of BTX-B that likely provides substantial binding energy (Fig. 3a, yellow). Finally, *D*III contributes C1343 and W1347 from the S5 segment to the binding site (Fig. 3a, yellow, Supplementary Fig. 6a, and Supplementary Movie 1, yellow sticks).

Thus, a total of 12 amino acid side chains from *D*III work together to form the core of Site IIB$_2$ (Fig. 3a, yellow).

In addition to these extensive interactions with *D*III, S1712 from the P-loop and I1758, I1759, and F1762 from the S6 segment in *D*IV all contribute to Site IIB$_2$ through interactions with the BTX-A core (Fig. 3a, red; Supplementary Fig. 6a; and Supplementary Movie 1, red sticks). The phenyl ring of F1762 makes a strong and well-elaborated hydrophobic interaction with the hydrophobic core of BTX-B, consistent with the strong negative impact on toxin binding and sodium channel activation for mutations at this site[24]. On the other hand, C899 and G900 are the only residues from *D*II in the P-loop that interact with BTX-B bound in Site IIB$_2$ through its benzoate moiety using their main chain backbone atoms rather than their side chains (Fig. 3a, upper left quadrant, green; Supplementary Fig. 6a; and Supplementary Movie 1, lime green sticks). Together with the extensive amino acid contacts with *D*III, these interactions with additional amino acid residues in *D*II and *D*IV make an exceptionally strong, specific, and three-dimensional Site IIB$_2$ for high-affinity binding of BTX-B.

As illustrated in Fig. 3b, the bound BTX-B at Site IIB$_1$ (bright green) nests in a cradle formed by hydrophobic residues from the *D*I P-loop and segment *D*I-S6 (Fig. 3b, blue), and segments *D*IV-S5 and *D*IV-S6 (Fig. 3b, red; Supplementary Fig. 6b; and Supplementary Movie 2). As highlighted in blue in Fig. 3b, the *D*I amino acid residue Q372 in the P-loop and M395, I398, F399, S402, and F403 in the *D*I-S6 segment form a key part of Site IIB$_1$ (Fig. 3b, blue; Supplementary Fig. 6b; and Supplementary Movie 2, blue sticks) through strong interaction with bound BTX-B (Fig. 3b, bright green). In addition to these extensive interactions with *D*I, several residues in *D*IV contribute to the binding of BTX-B to Site IIB$_1$ as highlighted in red (Fig. 3b, red; Supplementary Fig. 6b; and Supplementary Movie 2, red sticks). These amino acid residues in *D*IV include L1659, F1660, G1663, L1666, F1667 and M1670 from the *D*IV-S5 segment; F1707, T1710, and T1711 from the P-loop; and V1765, M1768, and Y1769 from the *D*IV-S6 segment (Fig. 3b, red, Supplementary Fig. 6b, and Supplementary Movie 2, red sticks). Altogether, these amino acid residues surround the bound BTX-B (bright

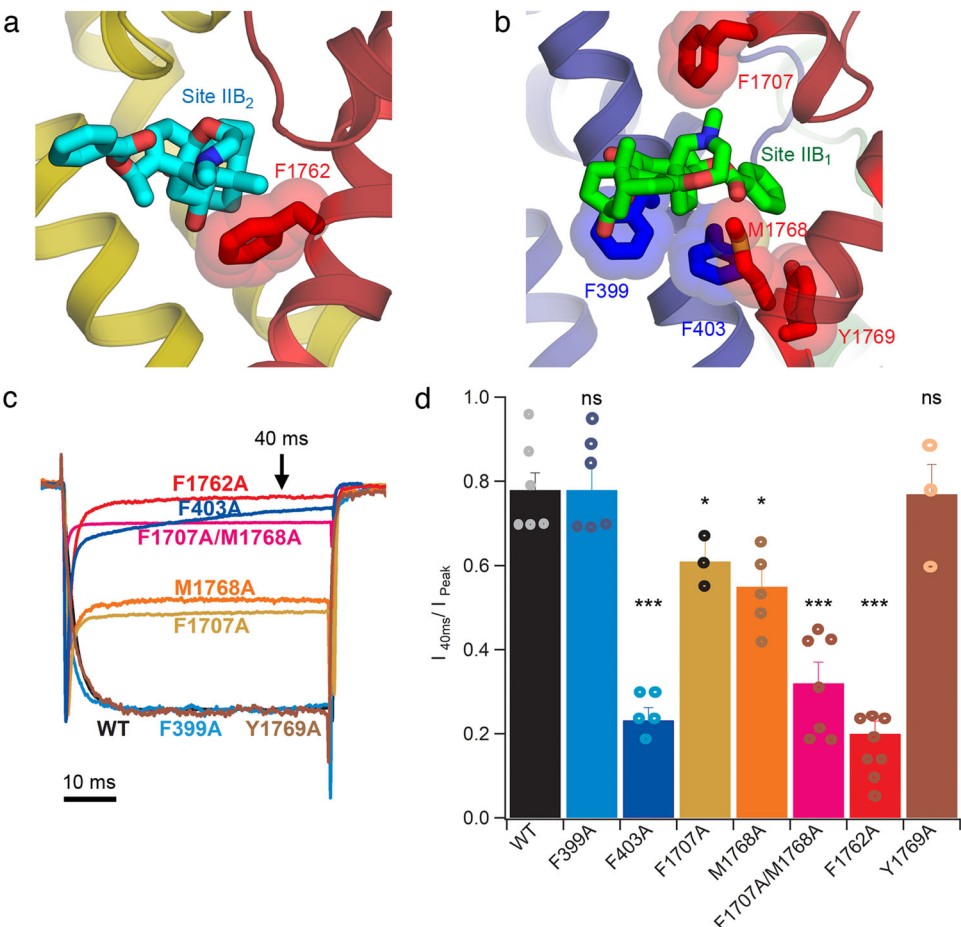

**Fig. 4 | Functional characterization of the dual receptor sites for BTX-B.**
**a** Location of F1762 (red stick) in *D*IV-S6 near Site IIB$_2$ (cyan sticks). **b** Locations of F399 and F403 (blue sticks) in *D*I-S6 and F1707, M1768, and Y1769 (red sticks) in *D*IV P-loop and S6 near Site IIB$_2$ (bright green sticks). **c** Raw traces of the peak current for rNa$_V$1.5c WT and mutants in the presence of 10 μM BTX-B. Raw traces for the mutants were scaled to the trace for the WT. Inward sodium current is plotted downward as a negative quantity, illustrating the rates and extents of voltage-dependent activation and inactivation. **d** Plot of inactivation ratios R$_i$ at 40 ms

following the peak for WT and mutants in the presence of 10 μM BTX-B. See Methods for details. Data are presented as mean values ± SEM with *n* = 6 cells for WT. Statistical significance was evaluated with Student's *t* test (two tailed test with no adjustment). Asterisks * and *** represent *p*-values ≤ 0.05 and ≤0.001, respectively; ns not significant. *p*-values: 0.972 (F399A, *n* = 6), 0.0000397 (F403A, *n* = 5), 0.0266 (F1707A, *n* = 3), 0.0130 (M1768A, *n* = 5), 0.0000845 (F1707A/M1768A, *n* = 7), 0.0000344 (F1762A, *n* = 8), and 0.903 (Y1769A, *n* = 3).

green sticks) and form a high-affinity three-dimensional Site IIB$_1$ for binding BTX-B.

The binding of BTX-B causes local induced-fit conformational changes in the rotamer of some key amino acid side chains. Notably, F399 and F403 from *D*I-S6, and F1667 from *D*IV-S5 adopt a "down" rotamer that allows for the binding of BTX-B in Site IIB$_1$ (Fig. 3b, blue: Supplementary Fig. 6b, and Supplementary Movie 2). F1467 from *D*III-S6 also adopts a new rotamer to interact with BTX-B in Site IIB$_2$ (Fig. 3a, yellow; Supplementary Figs. 5d and 6a, and Supplementary Movie 1). In the absence of the toxin, these residues adopt different rotamers in other Na$_V$1.5 structures that would either clash or make no interaction with BTX-B[11,46]. Further information on these rotamers in other Na$_V$1.5 structures is included in the legend to Fig. 3b. All amino acid residues forming the BTX-B receptor sites are conserved in human Na$_V$1.5 and many of them have clinical variants in which mutations are associated with cardiac arrhythmia (Supplementary Table 2).

### Functional characterization of dual BTX receptor sites
Extensive photoaffinity labeling and mutagenesis studies in *D*I, *D*III and *D*IV have shown that residues in these domains are critical for the binding and action of BTX-B[23–27]. These previous mutagenesis studies identified several individual residues observed in Site IIB$_2$ in our structure as part of the receptor site for BTX (see next section). On the

other hand, Site IIB$_1$ residues in *D*I were present in a peptide that was photoaffinity labeled by a photoreactive azido-BTX-B derivative[23] and the channels became BTX resistant when individual residues found in this photolabeled *D*I segment were mutated (see next section). However, Site IIB$_1$ contains some residues that have not been previously characterized by mutagenesis. To test whether the dual receptor sites observed in our cryo-EM structure of rNa$_V$1.5c are essential for the agonistic effect of BTX-B, we performed structure-guided mutagenesis and patch-clamp electrophysiological experiments with 10 μM BTX-B (Fig. 4 and Supplementary Figs. 8–11). A whole family of sodium current traces (−100 to 0 mV) were measured before and after bath application of BTX-B (Supplementary Figs. 8–11). At the depolarizing voltage that led to the biggest current amplitude, we measured the sodium current magnitude at 40 ms after the start of the depolarizing pulse and divided this value by the amplitude of the peak current in the same trace to obtain the inhibition ratio R$_i$ as a quantitative metric for the BTX-B action (Supplementary Figs. 10-11 and Supplementary Table 3) (see Methods). For the WT channel, the R$_i$ ratio was 0.78. As a positive control, we selected the *D*IV-S6 residue F1762 from Site IIB$_2$ for our initial mutagenesis study (Fig. 4a), as mutation of this residue has been shown to cause BTX resistance, and several other residues surrounding this site have already been characterized and validated for various Na$_V$ subtypes[24,26,36,47,48] (see next section). As expected, the

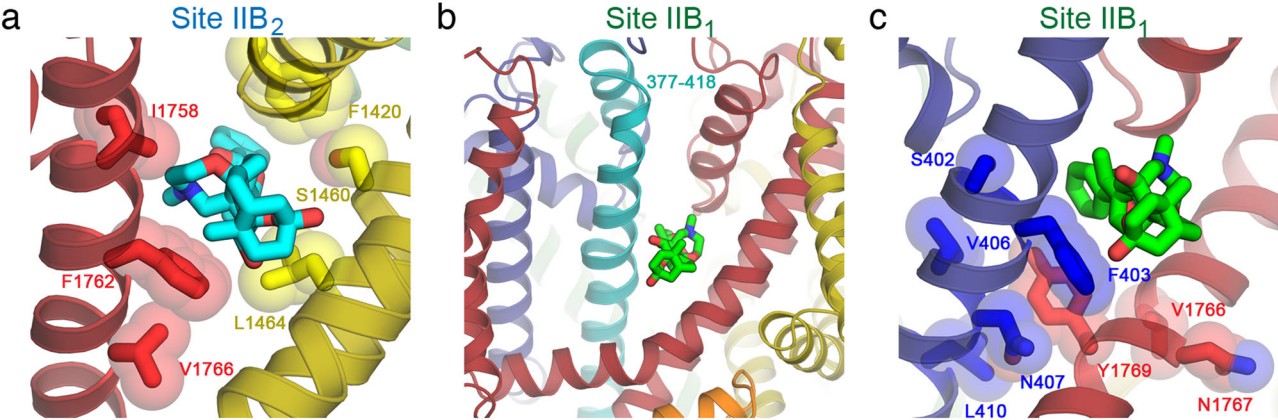

**Fig. 5 | Mapping of mutagenesis studies for BTX-B on the structure of rNa$_V$1.5c.**
**a** Residues near Site IIB$_2$ (cyan sticks) from *D*III (yellow) and *D*IV (red) that have been identified for BTX binding from mutagenesis studies[27,36,49]. The side chains are shown as sticks overlaid with transparent van der Waals spheres. **b** Location of residues N377 to E418 (teal) that includes *D*I-S6 of rNa$_V$1.5c near Site IIB$_1$ (bright green stick). This fragment is equivalent to residues N388 to E429 in rNa$_V$1.2 that has been shown to interact with azido BTX-B derivative by photoaffinity labeling[23]. **c** Residues near Site IIB$_1$ (bright green stick) from *D*I (blue) and *D*IV (red) that have been identified for BTX binding from mutagenesis studies[25,36,50] The side chains are shown as sticks overlaid with transparent van der Waals spheres. Y1769 is within 5 Å from the BTX-B but is not significantly involved in the binding based on mutagenesis studies.

F1762A showed a significant reduction of the functional effect of BTX-B by reducing the inhibition ratio R$_i$ to 0.20 ($p < 0.001$) (Fig. 4c, d). The F1762A mutation almost completely abolished the effect of BTX-B as the channel became nearly totally BTX-B resistant, consistent with results from previous studies[24]. As a negative control, we tested the *D*IV-S6 residue Y1769 (Fig. 4b), where mutations have been shown to have no effect on BTX function[24,26]. Consistent with previous studies, Y1769A did not show any reduction in the ratio R$_i$ from WT (Fig. 4c, d).

To evaluate Site IIB$_1$ further, residues F399 and F403 from the photolabeled segment in *D*I-S6, F1707 from *D*IV P-loop, and M1768 from *D*IV-S6 were selected (Fig. 4b). The F403A mutation caused a significant reduction of the BTX-B effect by reducing R$_i$ to 0.23 ($p < 0.001$) (Fig. 4c, d). The M1768A and F1707A mutations gave significantly smaller R$_i$ ratios of 0.55 ($p < 0.05$) and 0.61 ($p < 0.05$), respectively, compared to the value of 0.78 for the WT (Fig. 4 c, d), showing significant BTX-B resistance induced by these mutations. When we introduced the F1707A/M1768A double mutation to rNa$_V$1.5c, the R$_i$ ratio further decreased to 0.32 ($p < 0.001$), indicating a synergistic effect on BTX-B function between these two residues (Fig. 4c, d). The R$_i$ value for the F1707A/M1768A double mutations was similar to the R$_i$ value of 0.20 for the F1762A mutation in Site IIB$_2$. In contrast, the F399A mutation did not show any effect on BTX-B function, as the calculated R$_i$ ratio was 0.78, similar to the WT (Fig. 4c, d). It is possible that, due to its multiple rotamers, the bulky side chain of F399 may act as a gate to the fenestration rather than contributing directly to the BTX-B effect. Altogether, these results suggest that Site IIB$_1$ observed in our structure plays a significant role in the agonistic action of BTX-B.

### Structure/function analysis of dual BTX receptor sites
Our dual BTX-B receptor sites are well supported by previous mutagenesis and photoaffinity labeling studies, as summarized with colored shading of the numerous interacting components of these two binding sites with bound BTX-B in Fig. 5 and Supplementary Table 4. For Site IIB$_2$, contact residues F1420 in *D*III P-loop; S1460 and L1464 in *D*III-S6 (Fig. 5a, red); and I1758, F1762, and V1766 in *D*IV-S6 (Fig. 5a, yellow) are all implicated in BTX binding and action by mutagenesis studies in other Na$_V$ channels (Fig. 5a). F1236K and F1236R mutations in rNa$_V$1.4 (equivalent to F1420 in rNa$_V$1.5c) were also resistant to BTX[49] (Fig. 5a, yellow). S1276K and L1280K mutations in *D*III-S6 of rNa$_V$1.4 (equivalent to S1460 and L1464 in rNa$_V$1.5c) caused the channel to become BTX resistant[25,27] (Fig. 5a, yellow)). The I1760A mutation in *D*IV-S6 in Na$_V$1.2 (I1758 in rNa$_V$1.5c) caused the channel to have significantly reduced

sensitivity to BTX[24] (Fig. 5a, red). Mutations of residues equivalent to rNav1.5c F1762 in other channels (F1764A in Na$_V$1.2, F1710A and F1710I in Na$_V$1.3, and F1579K in rNa$_V$1.4) made them BTX resistant[24,26,36,48] (Fig. 5a, F1762, red), consistent with the F1762A mutation in our study (Fig. 3).

For Site IIB$_1$, the peptide segment N388 to Q429 in rat Na$_V$1.2 (corresponding to residues 377-418 in rNa$_V$1.5c) that included residues in *D*I-S6 (Fig. 5b, light blue) was found to be a binding site for BTX-B, as this segment of the channel was identified following peptide cleavage and antibody mapping to be specifically labeled by a photoreactive azido-derivative of BTX-B[23]. Mutation of S401 in hNa$_V$1.5 (S402 in rNa$_V$1.5c) to R or K prevented BTX action (Fig. 5c, blue), indicating that this position is close enough to the BTX binding site to allow steric or charge-charge clashes with the substituted R or K side chains, as mutation of this residue to other amino acids had no effect on BTX action[50]. The I433K, N434K, and L437K mutations in *D*I-S6 of rNa$_V$1.4 (V406, N407, and L410 in rNa$_V$1.5c) were BTX resistant while F430K (F403 in rNa$_V$1.5c) remained BTX sensitive[25,36] (Fig. 5c, blue). This is in contrast to the F403A mutation in our study that showed a significant BTX-B resistance (Fig. 4). For *D*IV-S6, BTX blocked cysteine modification in rNa$_V$1.4 with V1583C mutation (V1766 in rNav1.5c; Fig. 5a, c; red)[47]. N1584K and N1584A mutations in rNa$_V$1.4 (Asn1767 in rNa$_V$1.5c) were BTX resistant[26] (Fig. 5c, red). Interestingly, mutation of the residues equivalent to Y1769 in other sodium channels (Y1771A in Na$_V$1.2, and Y1586K and Y1586A in rNa$_V$1.4) had no effect, and the channels remained fully BTX sensitive similar to the Y1769A mutation in our study[24,26] (Fig. 5c, red). Altogether, these structure/function results summarized in Fig. 5 and Supplementary Table 4 further define the functional roles of a peptide segment in the *D*I-S6 segment and individual amino acid residues in *D*I, *D*III, and *D*IV in formation of BTX Site IIB$_1$ and Site IIB$_2$, providing strong support for the chemical interactions that we have observed at atomic resolution in our structure.

### Structural basis for agonistic actions of BTX-B
It has been suggested that S402 (*D*I-S6), N930 (*D*II-S6), S1460 (*D*III-S6), and F1762 (*D*IV-S6) form the SNSF pore-facing ring in the inner cavity of sodium channels that is near to or forms a part of the receptor sites for BTX-B (Fig. 6a)[50]. Each residue in the SNSF pore-facing ring is immediately adjacent to the flexible gating hinge G401, G929, G1459, and S1761 (GGGS) locus that is conserved among vertebrate sodium channels (Fig. 6a). Our structure shows that S402 directly forms an important interaction within Site IIB$_1$, whereas G1459, S1460 and F1762

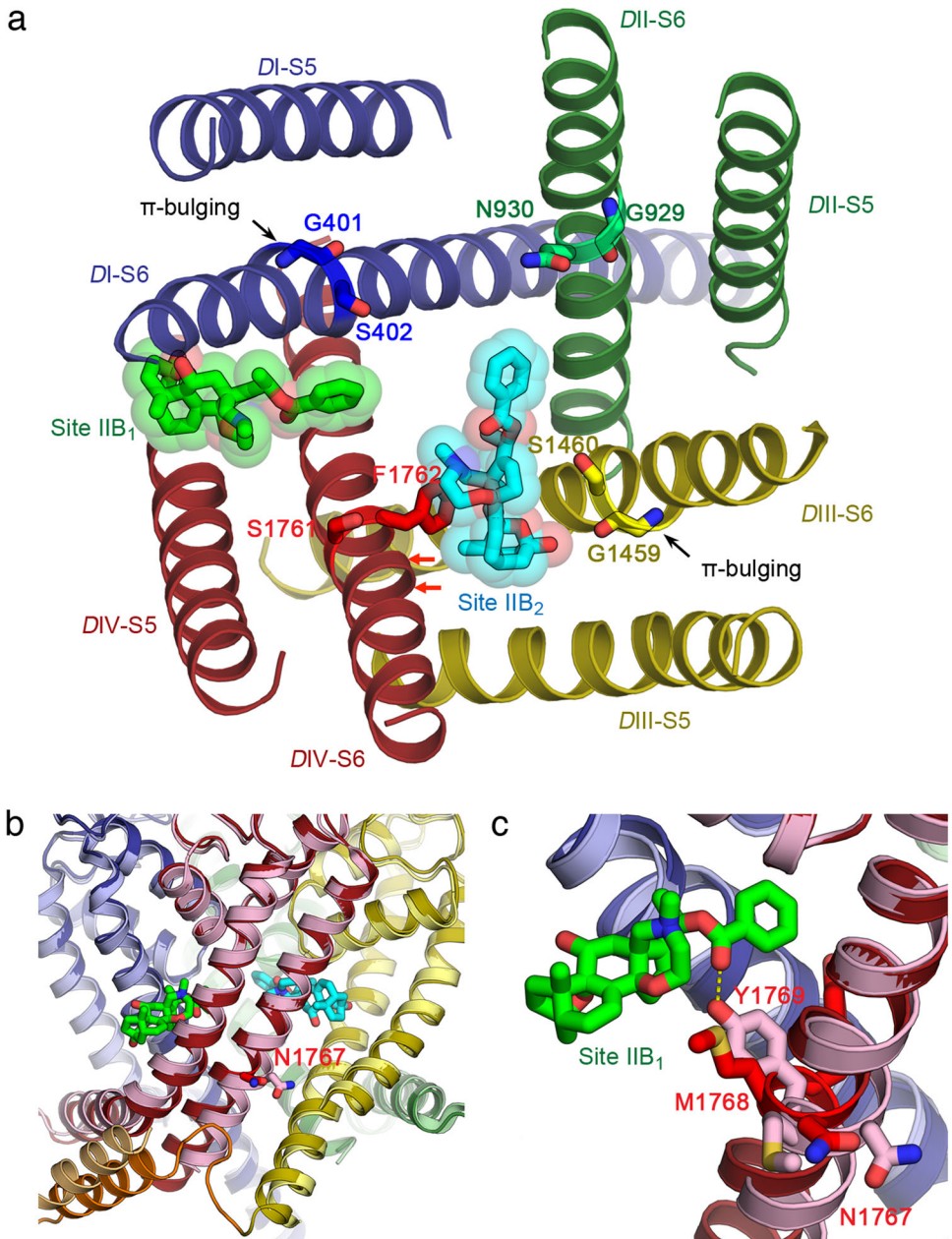

**Fig. 6 | Mechanism for the agonistic effects of BTX-B. a** The gating hinge and the π-helices in the structure of BTX-B bound rNaᵥ1.5c. Top view of the pore with only S5 and S6 segments shown for clarity. The gating hinge residues G401, G929, G1459, and S1761 on S6 from DI to DIV (GGGS motif) and their adjacent pore-facing residues S402, N930, S1460, and F1762 (SNSF motif) that have been shown to interact with Site II neurotoxins are shown as sticks colored according to their domains[26,27,50,52]. This section of S6 in DI and DIII adopt the π-helix configuration as indicated by the π-helix bulging while the section in DII and DIV are α-helix. BTX-B sites IIB₁ and IIB₂ are shown as sticks overlaid with transparent van der Waals sphere in cyan and bright green, respectively. Red arrows indicate the positions of I1758 and I1759. **b** The dual receptor sites for BTX-B are compatible with rNaᵥ1.5c in the open state. Superposition between the BTX-B bound rNaᵥ1.5c (DI−dark blue, DII−dark green, DIII−dark yellow, DIV−dark

red, DIII-DIV linker−orange) and the rNaᵥ1.5c QQQ in the Open state (DI−light blue, DII −light green, DIII−light yellow, DIV−pink, DIII-DIV linker−light orange) (PDB: 7FBS)[54]. Major conformational changes are observed in the C-terminal end of DIV-S6 (dark red vs. pink) starting at the conserved gating residue N1767 and in the DIII-DIV linker (dark orange vs. light orange) responsible for fast inactivation. The N1767 side chains from both structures are shown as sticks to indicate where the rotation of DIV-S6 starts. Site IIB₁ and IIB₂ are shown as bright green and cyan sticks, respectively. **c** Close-up view focusing on Site IIB₁ (bright green sticks). The rotation of the DIV-S6 from the BTX-B bound (red) to the open-state structure (pink) replaces the M1768 side chain with Y1769 that can hypothetically form a hydrogen bond with IIB₁ (yellow dash) to further stabilize the open state. However, Y1769 does not contribute significantly to the binding of BTX-B based on mutagenesis studies.

are part of Site IIB₂ (Figs. 3 and 6a). On the contrary, the N930 side chain is 6.5 Å away from the benzoate moiety of BTX-B in Site IIB₂ (Fig. 6a). When the corresponding residues in rat Naᵥ1.4 (N784) and human Naᵥ1.5 (N927) were mutated to K or R, the mutant channel became resistant to BTX, possibly due to adverse effects of the positive charge introduced with K or R side chains, because mutation to other amino acids had no effect[51,52]. Based on our analysis, the longer side

chain of N927K and N927R mutations in human Naᵥ1.5 (also N784K and L788K in rNaᵥ1.4) would bring this residue as close as 4 Å to Site IIB₂. This would limit the space in the pore and create charge repulsion that may block local conformational changes required for pore opening and may present a barrier for hydrated sodium ion to pass, while other mutations with a short side chain can still provide enough space in the cavity.

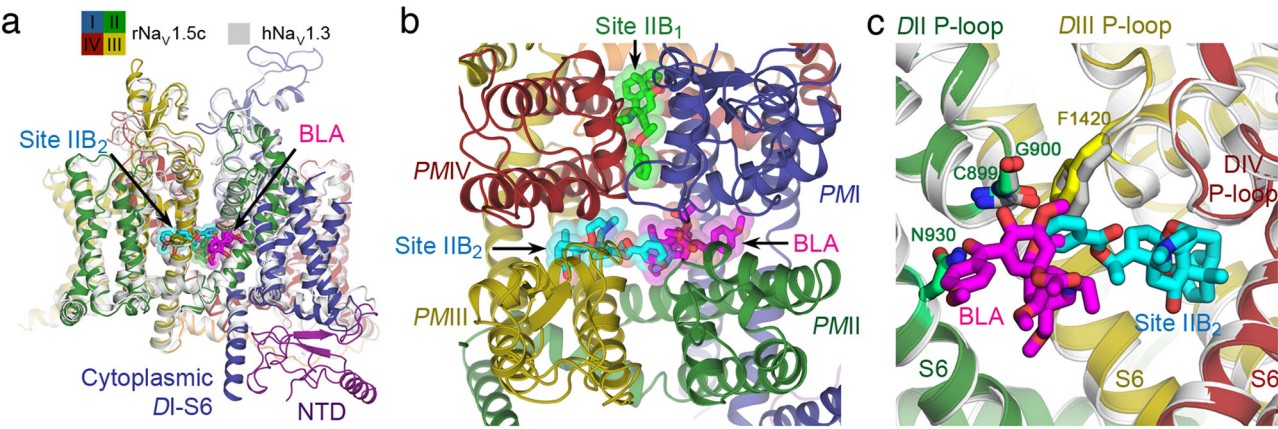

**Fig. 7 | Overlapping receptor sites for Neurotoxin Receptor Site II.**
**a** Superposition of BTX-B bound rNa$_V$1.5c with bulleyaconitine (BLA)-bound hNa$_V$1.3 (PDB: 7W77) (light gray) from the side view. Site IIB$_1$, Site IIB$_2$, and BLA are shown as sticks overlaid with transparent van der Waals spheres in bright green, cyan, and magenta, respectively. **b** Top view of Site IIB$_1$, Site IIB$_2$, and BLA. Site IIB$_2$

(cyan sticks) and BLA (magenta sticks) have an overlapping binding pose. **c** Close-up view of the superposition as in (A) showing the receptor sites for Site IIB$_2$ (cyan sticks) and BLA (magenta sticks). BTX-B and BLA compete for the same binding site near C899, G900 from *D*III (lime green) and F1420 from *D*III (yellow) in rNa$_V$1.5c. Equivalent side chains in hNa$_V$1.3 are shown as gray sticks.

The binding of BTX-B to both Site IIB$_1$ and Site IIB$_2$ appears to stabilize the π-helix conformation of the *D*I-S6 and *D*III-S6 segments, and the α-helical conformation in *D*II-S6 and *D*IV-S6 segments near the flexible gating hinge GGGS locus (Fig. 6a). In *D*I-S6, the π-bulging appears around G401 (Fig. 6a) to accommodate the recruitment of F399 and F403 for the binding at Site IIB$_1$ (Fig. 3b). In *D*III-S6, the π-bulging appears at G1459 (Fig. 6a), which provides the main chain oxygen to the binding of BTX-B in Site IIB$_2$ (Fig. 3a). On the other hand, the α-helix in *D*IV-S6 is stabilized by the binding of I1758, I1759, and F1762 to Site IIB$_2$ (Fig. 3a) adjacent to S1761 in the GGGS locus, and also by the binding of V1765, M1768 and Y1769 (Fig. 3b) near the conserved gating residue N1767. *D*II-S6 does not contribute to the binding of either BTX-B site and forms a continuous α-helix throughout the entire S6 segment. It is conceivable that this two-fold symmetric configuration of α-helix and π-helix in the four S6 segments allows BTX-B to stabilize an activated state of the channel without triggering fast inactivation.

Comparison to related structures of Na$_V$1.5 revealed that these π- and α-helix arrangements in the S6 segments of rNa$_V$1.5c/BTX-B are similar to those in some other structures. For *D*IV-S6, Site IIB$_2$ recruits F1762 (Fig. 6a, red) and nearby residues I1758 and I1759 (Fig. 6a, red arrows) for BTX-B binding and stabilizes the S6 conformation at the flexible gating hinge in the α-helical configuration. The α-helical form seen in this region is similar to the conformation found in other Na$_V$1.5 structures with the exception of the recently reported ranolazine-bound structure of rNa$_V$1.5c, which adopted the π-configuration instead[53]. The π-helix in *D*IV-S6 has also been seen in a fraction of the single particles in the recently reanalyzed structures of human Na$_V$1.7 with Site I and IV toxins bound[43]. The *D*IV-S6 π-to-α transition may be important for channel gating and can be modulated either directly or allosterically by drug and toxin ligands such as BTX-B to stabilize specific states of sodium channels.

We further examined whether the binding of BTX-B observed in our sodium channel structure captured in an inactivated state is compatible with the open-state conformation through which BTX-B asserts its agonistic effect. We compared our structure with the structure of the fast-inactivation-impaired rNa$_V$1.5c/QQQ (PDB: 7FBS), in which fast inactivation has been prevented by mutation of the essential IFM motif to QQQ (Fig. 6b, c)[40]. A major difference between the two structures occurs at the C-terminus of the *D*IV-S6 segment, starting at the conserved gating residue N1767, where the S6 α-helix begins to bend significantly away from the pore axis and widen the pore at the activation gate in the open-state structure (Fig. 6b, pink vs.

dark red). The *D*IV-S6 helix in the open state does not clash with BTX-B because the conformational change of the S6 segment mainly appears on the C-terminal side of the BTX-B receptor site (Fig. 6b). Our analysis revealed that both BTX-B sites are compatible with the open-state channel and require only changes in the backbone of flexible gating hinge regions and in the rotamers of a few side chains in the open state to avoid clashes. Only F1467 in *D*III-S6 in Site IIB$_2$ (Supplementary Fig. 7a, dark yellow vs. light yellow sticks) and F399 and F403 in *D*I-S6 in Site IIB$_1$ (Supplementary Fig. 7b, dark blue vs. light blue sticks), need to change their rotamers from "up" to "down" as seen in our structure to avoid clashing with BTX-B in the open state. The conformational change in the *D*IV-S6 segment in the open state shifted the amino acid register of residues on the C-terminal side of N1767 such that Y1769 displaces M1768 and can potentially form a hydrogen bond with BTX-B Site IIB$_1$ (Fig. 6c, pink) to stabilize the open state.

### BTX-B Site IIB$_2$ overlaps with other Site II neurotoxins
BTX-B is thought to have similar or overlapping binding sites with the other classic Site II neurotoxins, aconitine and veratridine, as demonstrated by ion flux and ligand competition experiments[34,35]. Currently only the structure of bulleyaconitine A (BLA), a well-characterized analog of aconitine, has been determined bound to human Na$_V$1.3 in an inactivated state (PDB: 7W77)[54]. In this structure, a single molecule of BLA is located in the fenestration between *D*I-*D*II of the PM, and its binding stabilized the helical conformation of the S6 segments at the flexible gating hinge as π-α-π-α from *D*I to *D*IV, similar to the configuration observed in our rNa$_V$1.5c/BTX-B structure. The structure of BLA-bound Na$_V$1.3 can be superimposed onto the BTX-B bound rNa$_V$1.5c structure with ~1.0 Å r.m.s.d. (Fig. 7a). BTX-B bound at Site IIB$_2$ extends its benzoate moiety toward Domain II and occupies part of the binding site for BLA (Fig. 7b), including the main chain of C899 and G900 from the *D*II P-loop, and the side chain of F1420 in the *D*III P-loop of rNa$_V$1.5c (Fig. 7c). This structural clash would create competitive binding interactions between BTX and aconitine, as observed in ion flux and ligand binding experiments[34,35]. It is possible that this clash might be more prominent in the activated open state of Na$_V$1.5, which would further impair persistent activation of sodium channels by BTX. Recently, a mutagenesis and computational modeling study has suggested that veratridine binds to rNa$_V$1.4 at a receptor site similar to Site IIB$_2$[55]. Our structural results reveal a potential mechanism for the competitive interactions of BTX with both of these structurally related Site II neurotoxins.

## Discussion

### Proposed mechanism of action of BTX

Binding of BTX favors activation of sodium channels by negatively shifting the voltage dependence of activation and preventing fast inactivation[28]. BTX-activated sodium channels also have altered ion conductance and selectivity[28]. The molecular and structural basis for these multifaceted toxin actions has been unknown. However, the locations of the dual BTX-B receptor sites at the interfaces of the *D*I-S6/*D*IV-S6 and *D*III-S6/*D*IV-S6 segments suggest probable structural mechanisms for these complex actions.

Previous structural studies showed that the *D*IV-S6 segment of $Na_V1.5$ bends, twists, and moves 6.1 Å away from the axis of the pore in order to open the activation gate at the intracellular ends of the S6 segments[40]. Shortly after opening of the pore, the fast inactivation gate folds into the PM, and the IFM inactivation particle binds in its receptor site formed by the intracellular end of the *D*IV-S6 segment and the S4-S5 linker adjacent to the activation gate, which pushes the activation gate into the closed conformation and induces fast inactivation[9,11,40]. Based on this previous structural work, we propose that the unique BTX-induced conformation of the *D*IV-S6 segment has two important effects. First, it favors bending the S6 segment toward its activated position by modifying the GGGS gating hinge through direct interactions with Site $IIB_1$ and Site $IIB_2$ and by inducing the unique π-α-π-α conformation of the S6 helices in Domains I-IV (Fig. 6a). Second, these S6 movements prevent fast inactivation by greatly slowing the binding of the IFM motif to its receptor site. BTX-B binding at Site $IIB_1$ also appears to stabilize the interaction between the cytoplasmic end of *D*I-S6 and the NTD allosterically. Since the NTD interacts with the *D*I-VS and forms a hydrogen bond with the *D*I S4-S5 linker in the activated state (Supplementary Fig. 3b), this may further contribute to energetic stabilization of the activated state.

Our structure suggests that BTX-B may cause a reduction of the single channel conductance by binding to Site $IIB_2$ and physically reducing the space in the central cavity of the pore, thereby reducing the rate of sodium influx. Moreover, BTX-B binding at Site $IIB_2$ also makes an apparent water-mediated hydrogen-bonding interaction with K1421 in the DEKA locus of the selectivity filter (Fig. 3a). Because K1421 is essential for sodium selectivity[56], toxin binding in this position likely modifies the interaction between K1421 and permeant ions, leading to further alterations in ion conductance and to physiologically important changes in ion selectivity. Site-directed mutations of K1421 greatly increase calcium permeation through $Na_V$ channels[56]. Therefore, the water-bridged hydrogen bond network formed by BTX-B and K1421 may be the source of the pathogenic calcium permeation through $Na_V$ channels induced by BTX. Together, these BTX actions would generate a modified open state with fast inactivation blocked, reduced sodium conductance, and greatly increased calcium selectivity.

Although this model for toxin action is consistent with the position of the BTX receptor sites, the intracellular ends of the S6 segments are interacting closely with each other and the conformation of the activation gate in our structure is tightly closed. Why has binding of BTX-B not opened the pore? Pore-opening requires movement of the intracellular ends of the S5 and S6 segments laterally, away from the axis of the pore. This movement is opposed by the force of the lateral pressure of the lipid bilayer, which is known to alter ion channel gating[57]. In a voltage-gated sodium channel, the S5 and S6 segments interact with this lateral force of the lipid bilayer directly and must overcome it to open the activation gate. We speculate that the lipid/detergent environment of our purified preparation of $rNa_V1.5c$ exerts a larger lateral pressure on the S6 segments than a native lipid bilayer and prevents the BTX-induced conformation of the activation gate and release of the IFM motif. Further structural studies with intact $Na_V1.5$ and more native lipid environment in nanodiscs or other protein/lipid preparations may allow these conformational movements to proceed and thereby trap the BTX-modified open state of the sodium channel for structure determination.

### Dual receptor sites for BTX

Our results reveal surprisingly that there are two receptor sites for BTX, here designated Site $IIB_1$ and Site $IIB_2$. Structure/function studies described above support the conclusion that both of these sites contribute to the functional effects of the bound toxin (Figs. 2–5). BTX has much higher affinity and efficacy in persistent activation of sodium channels than the other Site II toxins aconitine and veratridine[33,34], even though these toxins compete with each other for binding and action on sodium channels[35]. BTX is also far more toxic than aconitine and veratridine[15]. The dual binding sites for BTX-B observed in our structure may help to explain the strong agonistic effect of BTX.

Although such a model of dual receptor sites binding two small molecules of a toxin to a single molecular target is uncommon, the bivalent doubleknot toxin from the Earth Tiger tarantula has two peptide toxin moieties tethered in the single toxin molecule. It interacts with two identical target sites in the homotetrameric TRPV1 channel and irreversibly activates the channel[58]. The doubleknot structure of the toxin is required for high-affinity binding and irreversible activation of TRPV1[58]. These results provide clear precedent that binding of two toxin molecules can generate high-affinity binding and irreversible activation of an ion channel[59]. Another example is the Peruvian green velvet tarantula Protoxin-II, which binds to voltage sensors of $Na_V1.7$ in Domains II and IV, but no evidence for functional effects of toxin molecules bound at these two sites was presented[9]. Our studies show that binding of two small-molecule toxins to separate but homologous sites on a pseudotetrameric sodium channel can give both high affinity binding and irreversible activation and therefore support the proposal that these dual receptor sites underlie the exceptionally potent and multi-faceted effects of the toxin.

Past experiments have shown that BTX binding and action have a Hill coefficient near 1.0 (e.g. ref. 33), which suggests that there is a single binding site. However, if two binding sites have similar affinity and efficacy of toxin action, a Hill Coefficient near 1.0 may be expected. Moreover, the Hill Coefficient is defined for a process at equilibrium, whereas BTX action is irreversible on the time scale of biochemical and electrophysiological experiments in cellular preparations. Thus, these prior studies do not provide substantial evidence against dual BTX receptor sites.

### Fenestrations and BTX access

Fenestrations lead from the phospholipid bilayer into the central cavity of bacterial and mammalian sodium channels[5,9–11,60]. Their conservation over billions of years of evolution suggests an important biological function, which remains unknown. However, it is clear that fenestrations in the bacterial sodium channel $Na_VAb$ provide a pathway for pore-blocking local anesthetic and antiarrhythmic drugs to reach their receptor sites in the central cavity in the resting state[61]. In light of those results, it is intriguing that BTX-B binds within the fenestrations at Sites $IIB_1$ and $IIB_2$. We speculate that the fenestrations may provide an access pathway for BTX to reach these two subsites of Neurotoxin Receptor Site II in the resting state of sodium channels. It is conceivable that Site $IIB_1$ is an intermediate binding site along the fenestration for BTX en route into Site $IIB_2$ where BTX asserts its agonistic effect. However, our mutagenesis results for Site $IIB_1$ argue that this is not the case. Mutations of large amino acid side chains to Ala would have widened the fenestration to enhance access of ligands to the central cavity[61]. However, these mutations appeared to significantly diminish the functional effect of BTX-B instead of enhancing it, indicating that these side chains are important for the BTX-B function through significant contributions to toxin binding or to conformational stabilization.

## Multiple neurotoxin receptor sites on sodium channels

Our studies further define the multi-site model for toxin action on sodium channels at the atomic level[12,13,20–22,62]. The pore blockers tetrodotoxin, saxitoxin, and µ-conotoxins bind in overlapping positions in Neurotoxin Receptor Site I in the external vestibule and physically block the pore[9]. The polypeptide scorpion and sea anemone toxins bind to Neurotoxin Receptor Site III on the extracellular side of the VS in Domain IV and block functional coupling of VS activation to fast inactivation[22]. Similarly, the polypeptide toxins from spider venom bind to Neurotoxin Receptor Site IV and trap the VS in Domain II in its resting conformation, completely preventing the outward movement of the VS gating charges and the activation of the sodium channel[20,21]. BTX and other lipid-soluble toxins act differently, but ligand binding and ion flux studies led to the model that these toxins bind to a common site, designated Neurotoxin Receptor Site II, which enhances activation of sodium channels by an allosteric mechanism[12,34]. In contrast to expectation, the structural results presented here and in Li et al.[54] show that Site II toxins do not bind to a single binding site. Instead, BLA binds to a site at the interface of *PM*I and *PM*II (Fig. 6)[54], whereas we report here that BTX-B binds to two analogous sites at the interfaces between *PM*I and *PM*IV and between *PM*III and *PM*IV. BTX Site IIB$_2$ overlaps partially with the BLA binding site on *D*II (Fig. 7). This structural clash provides a molecular mechanism for competitive interactions between these toxins, even though their binding sites are closely spaced but not fully overlapping. In light of these results, we propose that the aconitine binding site be named Neurotoxin Site IIA, as it was first identified at the structural level, and that the BTX site be named Neurotoxin Receptor Site IIB. As we show here, Neurotoxin Receptor Site IIB is composed of two separate binding motifs, which we propose to name Site IIB$_1$ and Site IIB$_2$. These dual receptor sites provide a unique mechanism to enhance the potency and efficacy of BTX and greatly increase its toxicity to both predators and prey. In addition to revealing the underlying mechanism for the potent effects of BTX, our results provide a high-resolution template for structure-based design of sodium channel antagonist drugs that might occupy one of these sites, reduce hyperexcitability, and function as sodium channel inhibitors that would be useful in treatment of pain, cardiac arrhythmia, or epilepsy.

## Methods

### Microbe strains

*E. coli* GC10 (Genesee Scientific, catalog no. 42-661) was cultured at 37 °C in LB medium supplemented with 100 µg/mL of ampicillin for plasmid DNA extraction. *E. coli* DH10Bac (ThermoFisher Scientific, catalog no. 10361-012) was cultured at 37 °C in LB medium supplemented with 50 µg/mL kanamycin sulfate, 7 µg/mL gentamicin and 10 µg/mL tetracycline for bacmid production.

### Cell lines

Sf9 *(Spodoptera frugiperda)* insect cells (ThermoFisher Scientific, catalog no. B82501) were maintained in Grace's Insect Medium and supplemented with 10% FBS and penicillin/streptomycin at 27 °C and passaged at 80–95% confluence for baculovirus production. HEK293S GnTI⁻ *(Homo sapiens)* mammalian cells (American Type Culture Collection, catalog no. CRL-3022) were maintained and infected on cell culture plates in Dulbecco's Modified Eagle Medium (DMEM) supplemented with 10% FBS and glutamine/penicillin/streptomycin at 37 °C and 5% CO$_2$ for electrophysiology and protein expression. The cell lines were not authenticated or tested for Mycoplasma contamination.

### Mutagenesis

Site-directed mutagenesis of rNa$_V$1.5c mutants were generated by polymerase chain reaction (PCR) using Q5 High-Fidelity DNA polymerase (NEB) according to the manufacturer's protocol with pEG-BacMam rNa$_V$1.5c-eGFP-Flag plasmid[11,41] as a template. The 5′-phosphorylated oligonucleotides used as mutagenesis primers are listed in Supplementary Data. The resulting PCR products were treated with DpnI (NEB) at 37 °C overnight, followed by a PCR clean up (Omega Bio-tek). The purified DNAs were circularized using T4 DNA ligase (NEB) and transformed into *E. coli* GC10 chemically competent cells (Genesee Scientific). Plasmid DNAs were isolated from the cultures and mutagenesis results were confirmed by DNA sequencing. Baculoviruses were prepared using the Bac-to-Bac protocol from the plasmids according to the manufacturer with Sf9 insect cells (ThermoFisher Scientific).

### Electrophysiological recordings

All experiments were performed at room temperature (21–24 °C) as described previously[11]. rNa$_V$1.5c and its mutants were overexpressed in HEK293S GnTI⁻ cells using baculoviruses. Unless otherwise stated, HEK293S GnTI⁻ cells were held at −140 mV and 50-ms pulses were applied in 10 mV increments from −140 mV to +10 mV. A P/−4 holding leak potential was set to −140 mV. Extracellular solution contained in mM: 140 NaCl, 2 CaCl$_2$, 2 MgCl$_2$, 10 HEPES, pH 7.4. Intracellular solution contained: 35 NaCl, 105 CsF, 10 EGTA, 10 HEPES, pH 7.4. Glass electrodes had a resistance 1.5–2 MΩ. The pulse protocol was repeated every 5 min to assure that the effect of the 10 µM BTX-B had reached a plateau. Currents resulting from applied pulses were filtered at 5 kHz with a low-pass Bessel filter, and then digitized at 20 kHz. Data were acquired using an Axopatch 200B amplifier (Molecular Devices). Voltage commands were generated using Pulse 8.5 software (HEKA, Germany), and ITC18 analog-to-digital interface (Instrutech, Port Washington, NY).

### Analysis of electrophysiological data

Original records of examples of our electrophysiological analyses are presented in Supplementary Figs. 8–11. Current–voltage (I–V) relationships were recorded in response to voltage steps ranging from −140 to +10 mV in 10-mV increments from a holding potential of −140 mV. Conductance-voltage (G–V) curves were calculated from the corresponding (I–V) curves. Data were analyzed using Igor Pro 6.37 (WaveMetrics). G–V curves were fit with a Boltzmann Equation $1/(1+\exp(V_{1/2} - V_p)/k)$ in which $V_p$ is the stimulus potential, $V_{1/2}$ is the half-activation voltage, and $k$ is the slope factor. The data are presented as mean and standard error of the mean (SEM). Statistical significance was evaluated with Student's t-test.

To quantify the effect of BTX-B, we selected the pulse potential at 0 mV which gave maximum sodium current and complete steady state inactivation for all the mutants and the WT. We measured the sodium current amplitude at 40 ms after the start of the depolarizing pulse and divided this value by the amplitude of the peak current to obtain the inhibition ratio $R_i$ as a metric for BTX-B action. Because BTX-B slows the rate of inactivation, $R_i$ is large (~0.8) for WT with a strong toxin effect, whereas $R_i$ is much smaller (~0.2) for mutations that effectively block the toxin effect.

### Expression and purification of rNa$_V$1.5$_C$

Detailed expression of rNa$_V$1.5 $_C$ was described in our previous study[11,41]. Briefly, rNa$_V$1.5c was overexpressed in HEK293S GnTI⁻ cells using baculovirus generated from pEG-BacMam rNa$_V$1.5c-eGFP-Flag plasmid[11,41]. For purification, 12 L of infected cells were harvested and washed with Phosphate Buffer Saline (PBS) (Gibco) and 10% glycerol. Washed cell pellet was collected by centrifugation at $1000 \times g$ for 20 min and stored at −80 °C. The cell pellet was thawed and resuspended in 160 ml of PBS, supplemented with Roche cOmplete™ protease inhibitor cocktail (Sigma) and 600 nM BTX-B and lysed with Dounce homogenizer. The protein was extracted by mixing 160 mL of the cell lysate with 25 ml of stock solution containing n-dodecyl-β-D-maltopyranoside (DDM, Anatrace) and cholesteryl hemisuccinate Tris

salt (CHS, Anatrace) at the final concentration of 1% (w/v) DDM and 0.2% (w/v) CHS with agitation for 2 h at 4 °C. The supernatant was separated by ultra-centrifugation at 100,000 × $g$ for 30 min, then incubated with 5 mL of anti-Flag M2 affinity gel (Sigma) and agitated for 1 h at 4 °C. The resin was washed with 10 column volumes of Buffer A (25 mM HEPES pH = 7.4, 150 mM NaCl, 10% glycerol) supplemented with 0.06% glycoldiosgenin (GDN, Anatrace) and 600 nM BTX-B. The rNa$_V$1.5c protein was eluted with buffer A supplemented with 0.06% GDN, 600 nM BTX-B and 500 µg/mL Flag peptide (Bio Basic). Purified protein was subsequently loaded onto a Superose-6 column (GE Healthcare) pre-equilibrated in Buffer A, 0.06% GDN, and 600 nM BTX-B. Peak fractions were concentrated to ~1 mg/ml at 1 ml and mixed with 50 µg LqhIII (Latoxan Laboratory) and additional 12 nmol BTX-B. The mixture was then re-loaded to Superose-6 column pre-equilibrated with buffer containing 25 mM imidazole pH = 6.0, 150 mM NaCl, 0.006% GDN, and 600 nM BTX-B. Finally, peak fractions were concentrated to 60 µl at 5 mg/ml. Protein concentrations were estimated using 1 A$_{280}$ absorbance unit = 1 mg/ml on a NanoDrop spectrophotometer.

### Cryo-EM grid preparation and data collection
Three microliters of purified sample were applied to glow-discharged holey grids (QuantiFoil, 300 mesh, R1.2/1.3), and blotted for 2.5–4 s at 100% humidity and 22 °C before being plunge-frozen in liquid ethane cooled by liquid nitrogen using an FEI Mark IV Vitrobot. All data were acquired using a Titan Krios transmission electron microscope operated at 300 kV equipped with a Gatan K3 direct detector, and Gatan Quantum GIF energy filter with a slit width of 20 eV. A total of 7,542 movie stacks were automatically collected using SerialEM[63] at a nominal magnification of 105,000x with a pixel size of 0.41275 Å (super-resolution mode). Defocus range was set between −0.5 and −2 µm. Each stack was exposed for 2.04 s with 60 frames with a total dose of 60 e$^-$/Å$^2$.

### Cryo-EM data processing
CryoEM data were processed in CryoSPARC™ software systems version 3.3. The movie stacks were patch motion corrected, Fourier-cropped, and dose weighted, yielding a pixel size of 0.8255 Å. The contrast transfer function parameters for each motion-corrected image were estimated using patch CTF estimation in CryoSPARC. 7,063 micrographs with CTF fits better than 6 Å were used for particle picking. Approximately ~1.69 M particles were automatically picked using Blob Picker. After several rounds of 2D classification, ~558 K good particles were selected and subjected to 3D classification using the low-pass filtered cryoEM map of apo rNa$_V$1.5c[11] as the initial model. Heterogeneous refinement and non-uniform refinement protocols were employed for several iterations of 3D classification. After combining particles from the best 3D classes and removing duplicated particles, 86,763 particles were subjected to global and per-particle CTF refinement followed by non-uniform refinement and map sharpening. Global resolution estimation of the cryo-EM density map was based on the Fourier Shell Correlation criterion of 0.143. Local resolution was estimated in CryoSPARC. A diagram of data processing is presented in Supplementary Fig. 2.

### Model building and refinement
The structure of apo rNa$_V$1.5c α-subunit (PDB code: 6UZ3) was fitted into the cryo-EM density map of rNa$_V$1.5c in Chimera[64]. The NTD from the structure of human Na$_V$1.7 (PDB: 7XM9)[44], which is similar to the NTD structure predicted by AlphaFold for rNa$_V$1.5, was initially docked into the density map corresponding to the NTD region. The model was manually rebuilt in COOT[65] and subsequently refined using real-space refinement in Phenix[66]. The model vs. map FSC curve was calculated in Phenix. Statistics for cryo-EM data collection and model refinement are summarized in Supplementary Table 1.

### Reporting summary
Further information on research design is available in the Nature Portfolio Reporting Summary linked to this article.

## Data availability
The data that support this study are available from the corresponding authors upon request. The cryo-EM map has been deposited in the Electron Microscopy Data Bank (EMDB) under accession code EMD-41071 (the rNa$_V$1.5c/BTX-B complex). The atomic coordinates have been deposited in the Protein Data Bank (PDB) under accession code 8T6L (the rNa$_V$1.5c/BTX-B complex).

Additional structures used in this study:

3UZ0; 3UZ3; 6LQA; 7FBS; 7K18; 7W77; 7XM9. The source data underlying Figs. 1d and 4d, and Supplementary Fig. 1 are provided as a Source Data file. Requests for resources and reagents should be directed to and will be fulfilled by the Lead Contact, William A. Catterall (wcatt@uw.edu). All unique/stable reagents generated in this study are available from the Lead Contact with a completed Materials Transfer Agreement. Source data are provided with this paper.

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

## Acknowledgements

We thank Edelmar Navaluna (Department of Pharmacology, University of Washington) for technical support in baculovirus preparations, Dr. Jin Li (Department of Pharmacology, University of Washington) for technical support in molecular biology and for editorial support, and Catherine Garrison and Anne Wampler (Department of Chemistry, Stanford University) for technical support in chemical synthesis. This research was supported by National Institutes of Health Research Grants K08 HL145630 (M.J.L.), R35 NS111573 (W.A.C.), R01 HL112808 (W.A.C. and N.Z.), R01 GM117263 (J.D.), and by the Howard Hughes Medical Institute (N.Z.). A portion of this research was supported by National Institutes of Health Research Grant U24GM129547 and performed at the Pacific Northwest Center for Cryo-EM at Oregon Health & Science University, accessed through EMSL (grid.436923.9), a DOE Office of Science User Facility sponsored by the Office of Biological & Environmental Research.

## Author contributions

L.T., G.W., T.M.G., and W.A.C. conceived and designed experiments. L.T. purified protein and performed cryo-EM experiments with input from G.W. and M.J.L. L.T., G.W., and M.J.L. performed cryo-EM data analysis, atomic model building, and mutagenesis. T.M.G. performed electrophysiology experiments. M.M.L., T.T., and J.D. synthesized and supplied BTX-B. G.W., T.M.G., and W.A.C. analyzed the data and wrote the manuscript with input from L.T., M.J.L., J.D. and N.Z. All authors contributed to reviewing and revising the paper.

## Competing interests

J.D. is a cofounder and holds equity shares in SiteOne Therapeutics, Inc., a start-up company interested in developing subtype-selective modulators of Na$_V$s. The other authors declare no competing interests.
