## [Peer Review File · Nature Communications]

Dual receptor-sites reveal the structural basis for hyperactivation of sodium channels by poison-dart toxin batrachotoxinReviewer #1 (Remarks to the Author):

This study used a combination of cryogenic electron microscopy, site-directed mutagenesis of sodium channel cDNA, and electrophysiological analysis of heterologously expressed sodium channels to identify two homologous, but nonidentical receptor sites that simultaneously bind two molecules of the batrachotoxin derivative batrachotoxin-A 20-abenzoate (BTX-B). This unexpected result shows two binding sites, IIB1 and IIB2 -- one binding site at the interface between domains I and IV, and a second binding site at the interface between domains III and IV of the rat Nav1.5 sodium channel. The data suggest that these two bound BTX-B molecules stabilize the helical conformation of the S6 segments that gate the pore. One of the bound toxin molecules is shown to interact with residue Lys1421 that has been shown previously to be essential for sodium conductance and selectivity via an apparent water-bridged hydrogen bond. This work provides novel insights into the potency, efficacy, and functional effects of BTX-B on voltage-gated sodium channels via a dual receptor site mechanism. The high level of rigorous work represented in this manuscript is impressive. The results are in agreement with previous electrophysiological and biochemical data and, together, the work presents important structural and functional information on the interaction of BTX-B with Nav1.5. That said, there are some critical unanswered experimental questions. I also have suggestions to improve the manuscript, including the readability by and interest of non-experts in the field.

1. In the manuscript introduction, the lethal dose of BTX for mice is discussed. What is the lethal dose in humans?
2. Figure 3 tests the effects of a supramaximal BTX-B concentration (10 μ M) on sodium current inactivation for a series of Nav1.5 mutations. Do these mutations modulate the concentration-response relationship for BTX-B binding to Nav1.5? It would be more impactful to test a range of more physiologically relevant toxin concentrations. Does elimination of one site affect the affinity for the remaining site, and vice-versa? Are the two sites cooperative in their binding affinity and functional effects?
3. How do the Site IIB1 or Site IIB2 mutations tested in Figure 3 modulate channel selectivity for sodium vs. calcium?
4. How do the Site IIB1 or Site IIB2 mutations tested in Figure 3 modulate persistent (late) sodium current, which is a known BTX effect as well as a critical substrate for human cardiac arrhythmias?
5. The authors incorporated α -scorpion toxin (LqhIII) into their protein preparation to increase Nav1.5 affinity for BTX – but then do not detect the presence of LqhIII in the resulting structural data and make no further comments about this result. What does this mean? Why was LqhIII excluded from the channel represented in the cryo-em structure? Are there negatively cooperative interactions between Receptor Site(s) II and Receptor Site III in Nav1.5 such that when the BTX sites are filled, LqhIII can no longer be bound?
6. Families of current traces and activation/inactivation curves for each of the tested mutant channels in the absence and presence of BTX should be provided as Supplemental material. Presentation of these data would ensure that proper voltage control was achieved in all experiments.
7. Are BTX effects on Nav1.5 (or mutant Nav1.5) modulated by sodium channel beta subunits?
8. The authors used rat Nav1.5 cDNA/protein for their structural work. Are the identified toxin binding sites conserved in human Nav1.5? What about in the human neonatal vs. adult SCN5A alternative splice products? These questions should be addressed in the Discussion.
9. Also - are there documented human variants in SCN5A (e.g. identified in ClinVar or other databases) in the identified Site IIB1 or Site IIB2 regions?

10. The impact of the work could be strengthened by adding a section to the Discussion regarding how this new structural information might inform future drug discovery efforts.

11. The Results section contains a lot of information/commentary that should be moved to the Discussion (and some of this information is then repeated in the Discussion). The manuscript would be easier to read and understand if the Results section were just that – results – and all of the discussion and commentary moved to and incorporated in the Discussion section.

Minor comments:

1. Use of the word “unprecedented” in the abstract is a bit of an over-sell. I suggest instead the word “novel.”

Reviewer #2 (Remarks to the Author):

Voltage-gated sodium channels are crucial for action potential propagation in neurons and because of this are targets of many neurotoxins. One such neurotoxin is batrachotoxin (BTX) from poisonous frogs which hyperactivates NaVs, amongst other effects. The authors of the present study set out to structurally characterize the mode of action of BTX. Surprisingly, they find not one but two independent binding sites for BTX on NaV, a fact that explains some earlier observations. They describe the binding mode in detail and validate it by electrophysiological experiments. Their results help to understand the broad range of effects BTX has on NaV including inhibition of fast inactivation, loss of Na selectivity etc.

Overall, the study is very well executed and provides interesting and important insights for NaV regulation and can explain a lot of previous data on the mode of action of BTX. It is obvious that the authors are absolute experts in their field, with the last author being at the forefront of NaV research for the last decades. The manuscript breathes this expertise in that it is very focused on minuscule details; however, this level of detail and the in parts very technical structural descriptions make it somewhat inaccessible for non-experts. While I strongly support the publication of these data, the manuscript needs quite some rewriting to make it more accessible for the broad readership of Nature Communications by reducing the complexity where possible (e.g. by shortening or removing unnecessary sections like the description of bound annular lipids or the lengthy description of confirmatory mutations in other NaVs) and better introduce important principle like for example fast inactivation which might be a triviality for ion channel structural biologists but not necessarily for the non-experts.

Major points

- The introduction initiates with a lengthy description of the use of BTX as a dart poison. While I really enjoyed reading this section, it is not strictly necessary for understanding of the objective and I guess it could be condensed quite a bit in case it would be necessary to shorten the manuscript. However, I would not recommend this at this point. What I find more severe and not optimal is that the introduction starts directly into BTX and only later (and incompletely) introduces NaVs. This makes it very hard for readers who are not yet familiar with NaV channels. Furthermore, the introduction completely lacks a concise but complete introduction of fundamental and for the story important structural elements such as selectivity filter, voltage sensor, central cavity and intracellular gate, the mechanism of activation and fast inactivation and the fact that NaVs are pseudosymmetric. All of this should be included to allow the readers to understand and judge the results presented later.

- The description of BTX binding sites (p. 7-10) is obviously written by experts for experts. For someone not completely familiar with all the structural details of Nav, it will be very hard to follow and understand the binding mode. This section should be rewritten and described more systematically, by first generally locating the binding sites (and for example also mentioning that the two sites are not completely related to each other by pseudosymmetry), before going into some of the details.

- One example of unnecessary complexity is the lengthy description of rotamers of F399 and F403. While it might be interesting for absolute aficionados of NaV structures, the authors do not provide any functional relevance for rotamer change upon BTX binding (other than that it is necessary to prevent clashes), so it is not clear to me why that would need a whole paragraph. Better remove in order to not distract too much from the main story.

- For the experiment shown in Fig. 3c, please label the axes, especially to indicate where 40 ms (time point for the calculation for panel d) is. Also, it would be good to report the measured values for I40 ms, I_{Peak} and R_i in a supplementary table.

- The complete section "Structure/function analysis of dual BTX receptor sites" is not much more than a long list of mutations in other NaV channels which support the findings in this study. This is valuable information and impressively validates the observed binding sites, but could be condensed significantly since this level of detail is probably only relevant for absolute NaV-BTX experts but rather not so interesting for general readership. Maybe better to list these mutations in other NaV channels and their effects in a supplementary table instead of in the main text.

- In Fig. 6 B, it seems as if the binding sites of BLA and the IIB2 site of BTX-B might be very similar in a pseudo-symmetry-related manner. Is this the case or just an illusion from the figure? In case both binding sites are pseudo-symmetry related, would that give further insights into the modes of action of both neurotoxins?

Minor points:

- p. 5 last paragraph and Supplementary Fig. 1a: To call the SEC peak symmetrical is exaggerated, there is a clear shoulder on the left side plus it is not completely separated from the void peak. Please rephrase.

- p. 6 top, "1,238 amino acids spanning residue 11 in the NTD to residue 1,780...": The residue numbering is not completely clear to me, but I imagine that the 1,780 refers to the numbering of the full-length protein before truncating the intracellular linkers described above? Might be good to describe this unambiguously.

- Figure S3a does not really allow to judge the quality of fit of the NTD model into the density. It would be better to show only the NTD and omit the rest of the model, and show the density as mesh instead of as semi-transparent surface.

- The authors jump back and forth between one-letter and three-letter amino acid codes. I would suggest to stay consistent throughout the manuscript text and figures.

RESPONSE TO REVIEWER COMMENTS

We thank our reviewers for their positive review of our manuscript and for their detailed comments to improve it. The comments of our reviewers are copied below and our response to each comment is given below it in italics.

Reviewer #1 (Remarks to the Author):

This study used a combination of cryogenic electron microscopy, site-directed mutagenesis of sodium channel cDNA, and electrophysiological analysis of heterologously expressed sodium channels to identify two homologous, but nonidentical receptor sites that simultaneously bind two molecules of the batrachotoxin derivative batrachotoxin-A 20-abenzoate (BTX-B). This unexpected result shows two binding sites, IIB1 and IIB2 -- one binding site at the interface between domains I and IV, and a second binding site at the interface between domains III and IV of the rat Nav1.5 sodium channel. The data suggest that these two bound BTX-B molecules stabilize the helical conformation of the S6 segments that gate the pore. One of the bound toxin molecules is shown to interact with residue Lys1421 that has been shown previously to be essential for sodium conductance and selectivity via an apparent water-bridged hydrogen bond. This work provides novel insights into the potency, efficacy, and functional effects of BTX-B on voltage-gated sodium channels via a dual receptor site mechanism. The high level of rigorous work represented in this manuscript is impressive. The results are in agreement with previous electrophysiological and biochemical data and, together, the work presents important structural and functional information on the interaction of BTX-B with Nav1.5. That said, there are some critical unanswered experimental questions. I also have suggestions to improve the manuscript, including the readability by and interest of non-experts in the field.

1. In the manuscript introduction, the lethal dose of BTX for mice is discussed. What is the lethal dose in humans?

If it is assumed that human and mouse toxicity are roughly equivalent (at ~2.5 µg/kg injected subcutaneously), then a median lethal dose for a 68-kg (150 lb) human would be ~170 µg of BTX. Direct information is scarce because there are only rough estimates since few, if any, human poisonings have been reported in medical literature. (Dumbacher, J.P. Encyclopedia of Toxicology, 3rd Ed., 2014, pp. 371–373)

2. Figure 3 tests the effects of a supramaximal BTX-B concentration (10 µM) on sodium current inactivation for a series of Nav1.5 mutations. Do these mutations modulate the concentration-response relationship for BTX-B binding to Nav1.5? It would be more impactful to test a range of more physiologically relevant toxin concentrations. Does elimination of one site affect the affinity for the remaining site, and vice-versa? Are the two sites cooperative in their binding affinity and functional effects?

The concentration of BTX tested here is not a supramaximal dose under the conditions of these assays. The binding and action of BTX is highly use-dependent and voltage-dependent, which drives its high-affinity binding during repetitive firing of nerve and muscle. However, for equilibrium exposure in a cell culture setting, the EC₅₀ for persistent activation of sodium channels is 0.4 µM (Catterall, 1974). Thus, 10 µM is 25 times the EC₅₀ and should give about 97% occupancy. This level of saturation was selected so that effects of mutations would be easily detected. By measuring the % occupancy through fitting the inactivation time course, we

derive an accurate value for the apparent K_d , just as would be derived from a complete dose-response curve. We agree that a more complete analysis of binding interactions and cooperativity between the two BTX sites would be valuable; however, this analysis would require radiolabeled BTX to directly measure the number of sites occupied, which is not available.

3. How do the Site IIB1 or Site IIB2 mutations tested in Figure 3 modulate channel selectivity for sodium vs. calcium?

The effect of mutations in K1421 is well known--all sodium selectivity is lost whenever K1421 is mutated to any other amino acid residue because it is an integral part of the DEKA selectivity mechanism. The other mutations in the BTX sites would have little effect on ion selectivity because their side chains are in the central cavity and fenestrations rather than in the ion selectivity filter.

4. How do the Site IIB1 or Site IIB2 mutations tested in Figure 3 modulate persistent (late) sodium current, which is a known BTX effect as well as a critical substrate for human cardiac arrhythmias?

There is little or no effect of the mutations we have used on sustained sodium current, as shown in the current traces in Supplementary Figs. 8-11.

5. The authors incorporated a scorpion toxin (LqhIII) into their protein preparation to increase Nav1.5 affinity for BTX – but then do not detect the presence of LqhIII in the resulting structural data and make no further comments about this result. What does this mean? Why was LqhIII excluded from the channel represented in the cryo-em structure? Are there negatively cooperative interactions between Receptor Site(s) II and Receptor Site III in Nav1.5 such that when the BTX sites are filled, LqhIII can no longer be bound?

We were puzzled by this result as well. In short-term experiments, LqhIII and BTX were shown to act cooperatively. It is possible that, in the long time frame in the detergent solubilized state required for the cryo-EM study, the channel bound to BTX-B has become inactivated and caused the LqhIII to dissociate. Another possibility is that the EM particles corresponding to the full complex are too heterogeneous with respect to LqhIII binding to be separated into a single 3D class at high resolution.

6. Families of current traces and activation/inactivation curves for each of the tested mutant channels in the absence and presence of BTX should be provided as Supplemental material. Presentation of these data would ensure that proper voltage control was achieved in all experiments.

As requested, we have added Supplementary Figs. 8-11 comprehensively showing our current records for WT and mutants without and with BTX-B.

7. Are BTX effects on Nav1.5 (or mutant Nav1.5) modulated by sodium channel beta subunits?

The effects of sodium channel beta subunits on BTX actions have not been studied, in part because most of the electrophysiological studies of BTX action were carried out before cloning and expression of sodium channels was routine.

8. The authors used rat Nav1.5 cDNA/protein for their structural work. Are the identified toxin binding sites conserved in human Nav1.5? What about in the human neonatal vs. adult SCN5A alternative splice products? These questions should be addressed in the Discussion.

rNav1.5 and hNav1.5 are 94% conserved overall, and 100% conserved in the transmembrane segments of the pore where BTX-B sites are located (see Supplemental Table 2). The residues that form BTX-B sites in rNav1.5 appear to be identical in all human Nav channels.

The human neonatal and adult SCN5A spliced variants differ in the S3-S4 loop of VSD1. Because the pore domains are identical, the BTX-B sites are present in both spliced variants.

9. Also - are there documented human variants in SCN5A (e.g. identified in ClinVar or other databases) in the identified Site IIB1 or Site IIB2 regions?

Some residues identified as BTX-B sites in our structure have also been found in some human disease variants. We have summarized these clinical variants of human SCN5A in Supplementary Table 2.

10. The impact of the work could be strengthened by adding a section to the Discussion regarding how this new structural information might inform future drug discovery efforts.

Drugs that activate sodium channels as BTX does would be toxic due to their powerful nerve-muscle and cardiac effects. Drugs that act as inhibitors of sodium channels at the BTX sites might be of interest. We have added a sentence on this point at the end of the Discussion.

11. The Results section contains a lot of information/commentary that should be moved to the Discussion (and some of this information is then repeated in the Discussion). The manuscript would be easier to read and understand if the Results section were just that – results – and all of the discussion and commentary moved to and incorporated in the Discussion section.

Nature Communication discourages long Discussion sections and does not allow subheadings to identify different subsections, so it is not practical to move material from Results to Discussion. Moreover, in our opinion, comparison of our new structures with prior structures in the literature is appropriate for the Results sections of the manuscript because it is essential to support our discovery of dual receptor sites for BTX action.

Minor comments:

1. Use of the word “unprecedented” in the abstract is a bit of an over-sell. I suggest instead the word “novel.”

We have revised this to "novel" as requested. .

Reviewer #2 (Remarks to the Author):

Voltage-gated sodium channels are crucial for action potential propagation in neurons and because of this are targets of many neurotoxins. One such neurotoxin is batrachotoxin (BTX) from poisonous frogs which hyperactivates NaVs, amongst other effects. The authors of the present study set out to structurally characterize the mode of action of BTX. Surprisingly, they find not one but two independent binding sites for BTX on NaV, a fact that explains some earlier observations. They describe the binding mode in detail and validate it by electrophysiological experiments. Their results help to understand the broad range of effects BTX has on NaV including inhibition of fast inactivation, loss of Na selectivity etc.

Overall, the study is very well executed and provides interesting and important insights for NaV regulation and can explain a lot of previous data on the mode of action of BTX. It is obvious that the authors are absolute experts in their field, with the last author being at the forefront of NaV research for the last decades. The manuscript breathes this expertise in that it is very focused on minuscule details; however, this level of detail and the in parts very technical structural descriptions make it somewhat inaccessible for non-experts. While I strongly support the publication of these data, the manuscript needs quite some rewriting to make it more accessible for the broad readership of Nature Communications by reducing the complexity where possible (e.g. by shortening or removing unnecessary sections like the description of bound annular lipids or the lengthy description of confirmatory mutations in other NaVs) and better introduce important principle like for example fast inactivation which might be a triviality for ion channel structural biologists but not necessarily for the non-experts.

Major points

- The introduction initiates with a lengthy description of the use of BTX as a dart poison. While I really enjoyed reading this section, it is not strictly necessary for understanding of the objective and I guess it could be condensed quite a bit in case it would be necessary to shorten the manuscript. However, I would not recommend this at this point. What I find more severe and not optimal is that the introduction starts directly into BTX and only later (and incompletely) introduces NaVs. This makes it very hard for readers who are not yet familiar with NaV channels. Furthermore, the introduction completely lacks a concise but complete introduction of fundamental and for the story important structural elements such as selectivity filter, voltage sensor, central cavity and intracellular gate, the mechanism of activation and fast inactivation and the fact that NaVs are pseudosymmetric. All of this should be included to allow the readers to understand and judge the results presented later.

As requested, we have revised the Introduction and shortened it. The description of the structural components of the voltage sensor and pore domain of the sodium channel is now presented in Paragraph 1. We have retained a shortened description of the biology of BTX in Paragraph 2 to catch the interest of readers.

- The description of BTX binding sites (p. 7-10) is obviously written by experts for experts. For someone not completely familiar with all the structural details of Nav, it will be very hard to follow and understand the binding mode. This section should be rewritten and described more systematically, by first generally locating the binding sites (and for example also mentioning that the two sites are not completely related to each other by pseudosymmetry), before going into some of the details.

As requested, we have clarified our description of the amino acid residues that form the dual BTX-B binding sites. We have separated these results in a new Fig. 3 so that they can be displayed in larger format. In Fig. 3, we have color-coded the designations of amino acid residues using the same colors as the transmembrane segments where they are located. This allows us to designate these residues in color-coded groups as well as individual residues. In addition, we have summarized the residues forming BTX-B binding sites in Supplementary Table 2. With this new color scheme, the casual reader can easily see how the binding site is formed by the different transmembrane regions of Nav1.5, while the more in-depth reader can see the contribution of each amino acid residue in the same coloring scheme. We thank Reviewer II for bringing this problem to our attention, as we think that this new color-coding scheme now provides much clearer insights into the BTX-B receptor sites for the broad readership of Nature Communications.

- One example of unnecessary complexity is the lengthy description of rotamers of F399 and F403. While it might be interesting for absolute aficionados of NaV structures, the authors do not provide any functional relevance for rotamer change upon BTX binding (other than that it is necessary to prevent clashes), so it is not clear to me why that would need a whole paragraph. Better remove in order to not distract too much from the main story.

We have shortened and clarified our presentation of the rotamers of F399 and F403, but we need to keep the remaining basic information in the main text because the bound toxin does not fit its receptor site without the changes to the specific amino acid rotamers described.

- For the experiment shown in Fig. 3c, please label the axes, especially to indicate where 40 ms (time point for the calculation for panel d) is. Also, it would be good to report the measured values for I40 ms, IPeak and Ri in a supplementary table.

We have made these revisions and added Supplementary Figs. 8-11 to illustrate examples of our original recordings.

- The complete section “Structure/function analysis of dual BTX receptor sites” is not much more than a long list of mutations in other NaV channels which support the findings in this study. This is valuable information and impressively validates the observed binding sites, but could be condensed significantly since this level of detail is probably only relevant for absolute NaV-BTX experts but rather not so interesting for general readership. Maybe better to list these mutations in other NaV channels and their effects in a supplementary table instead of in the main text.

We have summarized mutations of residues forming BTX-B binding sites in other voltage-gated sodium channels in Supplementary Table 4, but we have kept an abbreviated form of this paragraph because it is of utmost importance for all readers to be convinced that we have correctly identified functionally significant dual receptor sites.

- In Fig. 6 B, it seems as if the binding sites of BLA and the IIB2 site of BTX-B might be very similar in a pseudo-symmetry-related manner. Is this the case or just an illusion from the figure? In case both binding sites are pseudo-symmetry related, would that give further insights into the modes of action of both neurotoxins?

The BLA site and the BTX-B site IIB₂ are somewhat homologous in a pseudosymmetric way. This is because they are located at the interface between two homologous pore modules that are opposite to each other (DI/DII in BLA vs. DIII/IV for BTX-B site IIB₂) at more or less the same height in the pore central cavity (at the fenestration level). They both mainly interact with residues in the S6 segments and in the bottom of the P-loops. However, this is where the similarity ends. Because these two compounds are quite different in the chemical structures and the pore modules from each domain are not identical, they make specific interactions with different amino acids in the channel (including some residues that are homologous). Importantly, Site IIB₂ includes strong interactions with K1421, which mediate the powerful effects of BTX on ion selectivity that are not part of the functional effects of BLA. We also note that BTX-B site IIB₂ lays mostly horizontal to the membrane plane while the core of BLA protrudes down toward the cytoplasmic ends of the S6 segments.

Minor points:

- p. 5 last paragraph and Supplementary Fig. 1a: To call the SEC peak symmetrical is exaggerated, there is a clear shoulder on the left side plus it is not completely separated from the void peak. Please rephrase.

We have removed the phrase “to yield a single symmetrical protein peak” from the sentence.

- p. 6 top, “1,238 amino acids spanning residue 11 in the NTD to residue 1,780...”: The residue numbering is not completely clear to me, but I imagine that the 1,780 refers to the numbering of the full-length protein before truncating the intracellular linkers described above? Might be good to describe this unambiguously.

The numbering is according to the WT full-length canonical sequence. We now highlight this important point in Page 4, Paragraph 2 before we begin the description of individual amino acid residues.

- Figure S3a does not really allow to judge the quality of fit of the NTD model into the density. It would be better to show only the NTD and omit the rest of the model, and show the density as mesh instead of as semi-transparent surface.

We have revised Supplementary Fig. S3 to include a new panel (panel b) showing the fit of the NTD model in the cryo-EM map in mesh.

- The authors jump back and forth between one-letter and three-letter amino acid codes. I would suggest to stay consistent throughout the manuscript text and figures.

As requested, we have revised to one-letter code, except in a few places where a one-letter code standing alone in the text could be confusing, such as A for Ala and I for Ile.

Reviewer #1 (Remarks to the Author):

The authors have done a nice job of revising their manuscript. I especially appreciate their efforts to make the writing more amenable to non-expert audiences. I have no further comments and recommend to accept the manuscript.

Reviewer #2 (Remarks to the Author):

I am satisfied with the authors' answers to the points I have raised earlier and suggest acceptance of the manuscript for publication.